# Joint Supervised and Self-supervised Learning for MRI Reconstruction

**George Yiasemis** [1,2] (ORCID)                      G.YIASEMIS@NKI.NL
**Nikita Moriakov** [1,2]                               N.MORIAKOV@NKI.NL
**Clara I. Sánchez** [2]                  C.I.SANCHEZGUTIERREZ@UVA.NL
**Jan-Jakob Sonke** [1,2]                                 J.SONKE@NKI.NL
**Jonas Teuwen** [1,2,3]                                 J.TEUWEN@NKI.NL
[1] *Netherlands Cancer Institute* [2] *University of Amsterdam* [3] *Radboud University Medical Center*

**Editors:** Accepted for publication at MIDL 2025

## Abstract

Magnetic Resonance Imaging (MRI) is a crucial imaging modality, but its inherently slow acquisition process limits the ability to obtain fully-sampled $k$-space data in motion-prone anatomical regions. The absence of such fully-sampled datasets, which serve as ground truths, hinders the supervised training of deep learning (DL) models—currently the state-of-the-art approach for MRI reconstruction. While self-supervised learning (SSL) methods attempt to overcome this limitation by training solely on subsampled $k$-space data, their performance remains inferior to supervised learning (SL). We propose Joint Supervised and Self-supervised Learning (JSSL), a novel training approach designed to enhance MRI reconstruction quality in cases where fully-sampled $k$-space data is unavailable for the *target* anatomy. JSSL jointly trains a model in a SSL setting using subsampled data from the target anatomy and in a SL manner using fully-sampled data from a *proxy* dataset, where full sampling is feasible. We evaluate JSSL on two distinct combinations of target and proxy datasets, demonstrating substantial improvements over conventional SSL methods through both quantitative and qualitative results. Additionally, we provide practical "rule-of-thumb" guidelines for selecting training strategies in MRI reconstruction. Our code is available at https://github.com/NKI-AI/direct.

**Keywords:** MRI Reconstruction, Inverse Problems, Deep Learning, SSL

## 1. Introduction

Magnetic Resonance Imaging (MRI) is widely used in clinical practice for its ability to non-invasively visualize detailed anatomical and physiological information. However, MRI data acquisition, known as $k$-space sampling, is inherently slow, increasing costs and limiting its feasibility in real-time applications such as MRI-guided radiotherapy. To accelerate scans, $k$-space subsampling techniques are employed, but they lead in lower-quality reconstructed images (Zbontar et al., 2019).

Deep learning (DL)-based MRI reconstruction techniques (Fessler, 2019; Pal and Rathi, 2022) have emerged as state-of-the-art solutions, outperforming conventional methods like Parallel Imaging (Pruessmann et al., 1999) and Compressed Sensing (Candes et al., 2006). These models are typically trained using supervised learning (SL), where retrospectively subsampled $k$-space data serve as inputs, and fully-sampled data act as ground truth.

Nevertheless, acquiring fully-sampled datasets, essential for SL training, is often infeasible or prohibitively expensive in certain anatomical regions, such as abdomen, cardiac cine, chest, or prostate imaging, where motion complicates adherence to the Nyquist-Shannon

sampling theorem (Shannon, 1949). Consequently, recent approaches have adopted self-supervised learning (SSL) strategies, which train DL-based algorithms using only subsampled $k$-space data, circumventing the need for fully-sampled ground truth.

A key SSL-based approach, Self-Supervised Learning via Data Undersampling (SSDU) (Yaman et al., 2020), proposed training reconstruction models by partitioning available subsampled $k$-space into input and target subsets, predicting one from the other. Inspired by Noise2Self (Batson and Royer, 2019), SSDU has since become a benchmark for SSL-based MRI reconstruction. Extensions include parallel networks (Hu et al., 2021), Noisier2Noise (Millard and Chiew, 2022) applying double subsampling, and architectures like Siamese networks with dual-domain loss functions (Yan et al., 2023) and coil sensitivity estimation (Hu et al., 2024; Millard and Chiew, 2022). These methods primarily rely on partitioning subsampled data, making SSDU a representative approach.

Other works combine supervised and self-supervised strategies. Noise2Recon (Desai et al., 2023) leverages both fully-sampled and subsampled data from the *same* domain for reconstruction and denoising, while (Zhou et al., 2022a) employs paired fully-sampled and subsampled data from *different* modalities, both still requiring fully-sampled data. Test-time training (Darestani et al., 2022) adapts pre-trained models to data shifts at inference via SSL-driven data-consistency loss, effective for domain shifts but impractical for real-time tasks due to re-training overhead. (A detailed review is provided in Appendix A.)

In this work, we propose Joint Supervised and Self-supervised Learning (JSSL), a novel method for training DL-based MRI reconstruction models without ground truth, fully-sampled $k$-space data for the target domain for SL. JSSL leverages fully-sampled data from *proxy* dataset(s) and subsampled data from *target* dataset(s) to jointly train models using both supervised and self-supervised paradigms, which reflects a realistic setting where in addition to the subsampled data for our clinical scenario some fully-sampled data from e.g. public reconstruction challenges are available. Our contributions include:

- JSSL presents the first approach to combine SL and SSL-based training in proxy and target organ domains within a unified pipeline for accelerated MRI reconstruction.
- We experimentally demonstrate that JSSL outperforms SSL-based MRI reconstruction approaches, with a specific focus on subsampled prostate and cardiac datasets.
- We offer practical "rule-of-thumb" guidelines for selecting appropriate training frameworks for accelerated MRI reconstruction models.

## 2. Materials and Methods

### 2.1. Introduction to MRI Acquisition and Reconstruction

A reconstructed image from subsampled multi-coil data $\tilde{\mathbf{y}}_{\mathbf{M}}$ can be obtained using the root-sum-of-squares (RSS) method: $\text{RSS} \circ \mathcal{F}^{-1}(\tilde{\mathbf{y}}_{\mathbf{M}}) := \left( \sum_{k=1}^{n_c} |\mathcal{F}^{-1}(\tilde{\mathbf{y}}_{\mathbf{M}}^k)|^2 \right)^{1/2}$. However, to recover a higher-quality image, a reconstruction is formulated as an optimization problem:

$$\mathbf{x}^* = \arg\min_{\mathbf{x}'} \frac{1}{2} \left\| \mathcal{A}_{\mathbf{M},\mathbf{S}}(\mathbf{x}') - \tilde{\mathbf{y}}_{\mathbf{M}} \right\|_2^2 + \mathcal{G}(\mathbf{x}'), \tag{1}$$

where $\mathcal{G}$ represents an arbitrary regularization functional, and $\mathcal{A}_{\mathbf{M},\mathbf{S}} : \mathbb{C}^n \to \mathbb{C}^{n \times n_c}$ denotes the forward operator, which sequentially applies the coil expansion operator $\mathcal{E}_{\mathbf{S}}$, the Fourier transform $\mathcal{F}$, and a subsampling mask operator $\mathbf{U}_{\mathbf{M}}$:

$$\mathcal{A}_{\mathbf{M},\mathbf{S}}(\mathbf{x}) = \mathbf{U}_{\mathbf{M}} \circ \mathcal{F} \circ \mathcal{E}_{\mathbf{S}}(\mathbf{x}) = \left\{ \mathbf{U}_{\mathbf{M}} \circ \mathcal{F}(\mathbf{S}^k \mathbf{x}) \right\}_{k=1}^{n_c}, \quad \mathbf{U}_{\mathbf{M}}(\mathbf{y})_j = \mathbf{y}_j \cdot \mathbb{1}_{\mathbf{M}}(j). \tag{2}$$

Equation (1) is typically solved using unrolled optimization methods (Yiasemis et al., 2022b), which leverage both the forward operator $\mathcal{A}_{\mathbf{M},\mathbf{S}}$ and its adjoint $\mathcal{A}_{\mathbf{M},\mathbf{S}}^* : \mathbb{C}^{n \times n_c} \to \mathbb{C}^n$. The adjoint operator reconstructs an image by applying subsampling via $\mathbf{U}_{\mathbf{M}}$, inverse Fourier transform $\mathcal{F}^{-1}$, and coil combination through the reduction operator $\mathcal{R}_{\mathbf{S}}$:

$$\mathcal{A}_{\mathbf{M},\mathbf{S}}^*(\mathbf{y}) = \mathcal{R}_{\mathbf{S}} \circ \mathcal{F}^{-1} \circ \mathbf{U}_{\mathbf{M}}(\mathbf{y}) = \sum_{k=1}^{n_c} \mathbf{S}^{k*} \mathcal{F}^{-1} \left( \mathbf{U}_{\mathbf{M}}(\mathbf{y}^k) \right). \tag{3}$$

Deep learning (DL)-based approaches eliminate the need for explicit regularization terms, instead learning the reconstruction directly from data (Singh et al., 2023).

## 2.2. MRI Reconstruction with Supervised Learning

In supervised learning settings, fully-sampled $k$-space datasets are assumed to be available. Let $\mathcal{D}^{\text{SL}} = \{\mathbf{y}^{(i)}\}_{i=1}^N$ represent such a dataset, which is retrospectively subsampled during training: $\tilde{\mathbf{y}}_{\mathbf{M}_i}^{(i)} = \mathbf{U}_{\mathbf{M}_i}(\mathbf{y}^{(i)})$, and let $f_{\boldsymbol{\psi}}$ denote a DL-based reconstruction network with parameters $\boldsymbol{\psi}$. Note that the architecture of $f_{\boldsymbol{\psi}}$ can be configured to output image reconstructions, $k$-space data, or both, but here we assume that both input and output lie in the image domain. The objective in SL-based MRI reconstruction is to minimize the discrepancy between the fully-sampled and predicted $k$-spaces:

$$\boldsymbol{\psi}^* = \arg\min_{\boldsymbol{\psi}} \frac{1}{N} \sum_{i=1}^N \mathcal{L}_K \left( \mathbf{y}^{(i)}, \hat{\mathbf{y}}^{(i)} \right), \ \hat{\mathbf{y}}^{(i)} = \text{DC}_{\mathbf{M}_i} \left( \tilde{\mathbf{y}}_{\mathbf{M}_i}^{(i)}, \mathcal{F} \circ \mathcal{E}_{\mathbf{S}} \circ f_{\boldsymbol{\psi}} \left( \tilde{\mathbf{x}}_{\mathbf{M}_i}^{(i)} \right) \right), \ \tilde{\mathbf{x}}_{\mathbf{M}_i}^{(i)} = \mathcal{A}_{\mathbf{M}_i,\mathbf{S}}^* \left( \mathbf{y}^{(i)} \right). \tag{4}$$

Here, $\text{DC}_{\mathbf{M}}$ denotes the data consistency operator which ensures that the reconstructed data remain consistent with the available measurements: $\text{DC}_{\mathbf{M}}(\mathbf{w}_1, \mathbf{w}_2) = \mathbf{U}_{\mathbf{M}}(\mathbf{w}_1) + \mathbf{U}_{\mathbf{M}^c}(\mathbf{w}_2)$. Loss can also minimize the discrepancy between the fully-sampled and predicted images:

$$\boldsymbol{\psi}^* = \arg\min_{\boldsymbol{\psi}} \frac{1}{N} \sum_{i=1}^N \mathcal{L}_{\text{I}} \left( \mathbf{x}^{(i)}, \hat{\mathbf{x}}^{(i)} \right), \quad \mathbf{x}^{(i)} = \text{RSS} \circ \mathcal{F}^{-1} \left( \mathbf{y}^{(i)} \right), \quad \hat{\mathbf{x}}^{(i)} = \left| f_{\boldsymbol{\psi}} \left( \tilde{\mathbf{x}}_{\mathbf{M}_i}^{(i)} \right) \right|, \tag{5}$$

where $\mathcal{L}_K$ and $\mathcal{L}_{\text{I}}$ denote arbitrary frequency and image domain loss functions. Although effective, SL methods depend on fully sampled data, which may not always be available.

## 2.3. MRI Reconstruction with Self-supervised Learning

When fully-sampled $k$-space data are unavailable, DL models can be trained using SSL. Let $\mathcal{D}^{\text{SSL}} = \{\tilde{\mathbf{y}}_{\mathbf{M}_i}^{(i)}\}_{i=1}^N$ represent a dataset of subsampled acquisitions, where each instance $\tilde{\mathbf{y}}_{\mathbf{M}_i}^{(i)}$ is sampled from a set $\mathbf{M}_i$. In SSL, training involves partitioning the acquired subsampled measurements (Yaman et al., 2020). For each sample $\tilde{\mathbf{y}}_{\mathbf{M}_i}^{(i)}$, the sampling pattern $\mathbf{M}_i$ is divided into two disjoint subsets, $\boldsymbol{\Theta}_i$ and $\boldsymbol{\Lambda}_i$, followed by projecting $\tilde{\mathbf{y}}_{\mathbf{M}_i}^{(i)}$ onto both:

$$\boldsymbol{\Theta}_i \cup \boldsymbol{\Lambda}_i = \mathbf{M}_i, \quad \boldsymbol{\Theta}_i \cap \boldsymbol{\Lambda}_i = \emptyset, \quad \tilde{\mathbf{y}}_{\boldsymbol{\Theta}_i}^{(i)} = \mathbf{U}_{\boldsymbol{\Theta}_i}(\tilde{\mathbf{y}}_{\mathbf{M}_i}^{(i)}), \quad \tilde{\mathbf{y}}_{\boldsymbol{\Lambda}_i}^{(i)} = \mathbf{U}_{\boldsymbol{\Lambda}_i}(\tilde{\mathbf{y}}_{\mathbf{M}_i}^{(i)}). \tag{6}$$

Subsequently, one partition $(\tilde{\mathbf{y}}_{\boldsymbol{\Lambda}_i}^{(i)})$ is used as input to the reconstruction network, while the other serves as the target $(\tilde{\mathbf{y}}_{\boldsymbol{\Theta}_i}^{(i)})$. The objective loss function is formulated in the $k$-space domain minimizing discrepancy between the target k-space partition and the predicted k-space restricted on the target partition $(\hat{\mathbf{y}}_{\boldsymbol{\Theta}_i \boldsymbol{\Lambda}_i}^{(i)})$:

$$\boldsymbol{\psi}^* = \arg\min_{\boldsymbol{\psi}} \frac{1}{N} \sum_{i=1}^{N} \mathcal{L}_K(\tilde{\mathbf{y}}_{\boldsymbol{\Theta}_i}^{(i)}, \hat{\mathbf{y}}_{\boldsymbol{\Theta}_i \boldsymbol{\Lambda}_i}^{(i)}),$$

$$\hat{\mathbf{y}}_{\boldsymbol{\Theta}_i \boldsymbol{\Lambda}_i}^{(i)} = \mathbf{U}_{\boldsymbol{\Theta}_i}\left(\mathrm{DC}_{\boldsymbol{\Lambda}_i}\left(\tilde{\mathbf{y}}_{\boldsymbol{\Lambda}_i}^{(i)}, \hat{\mathbf{y}}_{\boldsymbol{\Lambda}_i}^{(i)}\right)\right), \quad \hat{\mathbf{y}}_{\boldsymbol{\Lambda}_i}^{(i)} = \mathcal{F} \circ \mathcal{E}_{\mathbf{S}} \circ f_{\boldsymbol{\psi}}(\tilde{\mathbf{x}}_{\boldsymbol{\Lambda}_i}^{(i)}), \quad \tilde{\mathbf{x}}_{\boldsymbol{\Lambda}_i}^{(i)} = \mathcal{A}_{\boldsymbol{\Lambda}_i, \mathbf{S}}^*(\tilde{\mathbf{y}}_{\mathbf{M}_i}^{(i)}). \tag{7}$$

While most SSL-based MRI reconstruction methods rely on loss calculations in the frequency domain (Yaman et al., 2020; Millard and Chiew, 2022; Hu et al., 2021, 2024), some studies have explored dual-domain losses (Zhou et al., 2022b; Yan et al., 2023). The loss can equivalently be computed in the image domain as follows:

$$\boldsymbol{\psi}^* = \arg\min_{\boldsymbol{\psi}} \frac{1}{N} \sum_{i=1}^{N} \mathcal{L}_{\mathrm{I}}\left(\tilde{\mathbf{x}}_{\boldsymbol{\Theta}_i}^{(i)}, \hat{\mathbf{x}}^{(i)}\right), \quad \tilde{\mathbf{x}}_{\boldsymbol{\Theta}_i}^{(i)} = \mathrm{RSS} \circ \mathcal{F}^{-1}\left(\tilde{\mathbf{y}}_{\boldsymbol{\Theta}_i}^{(i)}\right), \quad \hat{\mathbf{x}}^{(i)} = \left|\mathcal{R}_{\mathbf{S}} \circ \mathcal{F}^{-1}\left(\hat{\mathbf{y}}_{\boldsymbol{\Theta}_i \boldsymbol{\Lambda}_i}^{(i)}\right)\right|. \tag{8}$$

SSL-based MRI reconstruction thus learns by partitioning acquired measurements, using one subset as input and the other as the training target. Although SSL reduces reliance on fully-sampled data, it often underperforms compared to SL.

## 2.4. Joint Supervised and Self-supervised Learning

To address the limitations of SSL in scenarios where fully-sampled data are unavailable in the target domain, we propose Joint Supervised and Self-supervised Learning. JSSL integrates SSL using subsampled measurements from target domain(s) with SL using fully-sampled acquisitions from proxy datasets in other organ domains. By leveraging knowledge from proxy datasets, JSSL aims to surpass the performance of conventional SSL methods that rely solely on subsampled target data. Figure 1 illustrates the end-to-end JSSL pipeline. A theoretical rationale for JSSL is provided in Appendix B, where we argue that training on a proxy task could reduce estimator error by reducing estimator variance while introducing a negligible bias.

**JSSL Training Framework**   To implement JSSL, we construct the overall loss function with two components: one for supervised learning and another for self-supervised learning. For simplicity we assume a single target and a single proxy dataset in our definitions.

**Supervised Learning Loss**   The SL loss is calculated on the proxy dataset, which contains fully-sampled $k$-space data. It is formulated as follows:

$$\mathcal{L}_{\boldsymbol{\psi}}^{\mathrm{SL}} := \mathcal{L}_{\mathrm{I}\boldsymbol{\psi}}^{\mathrm{SL}} + \mathcal{L}_{K\boldsymbol{\psi}}^{\mathrm{SL}} = \frac{1}{N_{\mathrm{prx}}} \sum_{i=1}^{N_{\mathrm{prx}}} \left[\mathcal{L}_{\mathrm{I}}\left(\mathbf{x}^{\mathrm{prx},(i)}, \hat{\mathbf{x}}^{\mathrm{prx},(i)}\right) + \mathcal{L}_K\left(\mathbf{y}^{\mathrm{prx},(i)}, \hat{\mathbf{y}}^{\mathrm{prx},(i)}\right)\right]. \tag{9}$$

Here, $\mathbf{x}^{\mathrm{prx},(i)}$ , $\hat{\mathbf{x}}^{\mathrm{prx},(i)}$ represent the ground truth and predicted images, respectively, for the $i$-th sample in the proxy dataset, while $\mathbf{y}^{\mathrm{prx},(i)}$, $\hat{\mathbf{y}}^{\mathrm{prx},(i)}$ represent the fully-sampled and predicted $k$-spaces, respectively, as defined in Sec. 2.2.

**Self-supervised Learning Loss**   The SSL loss is calculated using the target dataset, consisting of subsampled $k$-space data without ground truth. Motivated by SSL-based methods (Zhou et al., 2022a,b) which established improved performance when using dual-domain loss, we calculate the SSL loss in both the image and $k$-space domains as follows:

$$\mathcal{L}_{\boldsymbol{\psi}}^{\mathrm{SSL}} := \mathcal{L}_{\mathrm{I}\boldsymbol{\psi}}^{\mathrm{SSL}} + \mathcal{L}_{K\boldsymbol{\psi}}^{\mathrm{SSL}} = \frac{1}{N_{\mathrm{tar}}} \sum_{i=1}^{N_{\mathrm{tar}}} \left[\mathcal{L}_K\left(\tilde{\mathbf{y}}_{\boldsymbol{\Theta}_i}^{\mathrm{tar},(i)}, \hat{\mathbf{y}}_{\boldsymbol{\Theta}_i \boldsymbol{\Lambda}_i}^{\mathrm{tar},(i)}\right) + \mathcal{L}_{\mathrm{I}}\left(\tilde{\mathbf{x}}_{\boldsymbol{\Theta}_i}^{\mathrm{tar},(i)}, \hat{\mathbf{x}}^{\mathrm{tar},(i)}\right)\right], \tag{10}$$

$$\tilde{\mathbf{x}}_{\boldsymbol{\Theta}_i}^{\mathrm{tar},(i)} = \mathrm{RSS} \circ \mathcal{F}^{-1}(\tilde{\mathbf{y}}_{\boldsymbol{\Theta}_i}^{\mathrm{tar},(i)}), \quad \hat{\mathbf{x}}^{\mathrm{tar},(i)} = \left|f_{\boldsymbol{\psi}}(\tilde{\mathbf{x}}_{\boldsymbol{\Lambda}_i}^{\mathrm{tar},(i)})\right|,$$

where, $\tilde{\mathbf{x}}_{\boldsymbol{\Lambda}_i}^{\mathrm{tar},(i)}$, $\tilde{\mathbf{y}}_{\boldsymbol{\Theta}_i}^{\mathrm{tar},(i)}$, $\hat{\mathbf{y}}_{\boldsymbol{\Theta}_i \boldsymbol{\Lambda}_i}^{\mathrm{tar},(i)}$ are as defined in Sec. 2.3.

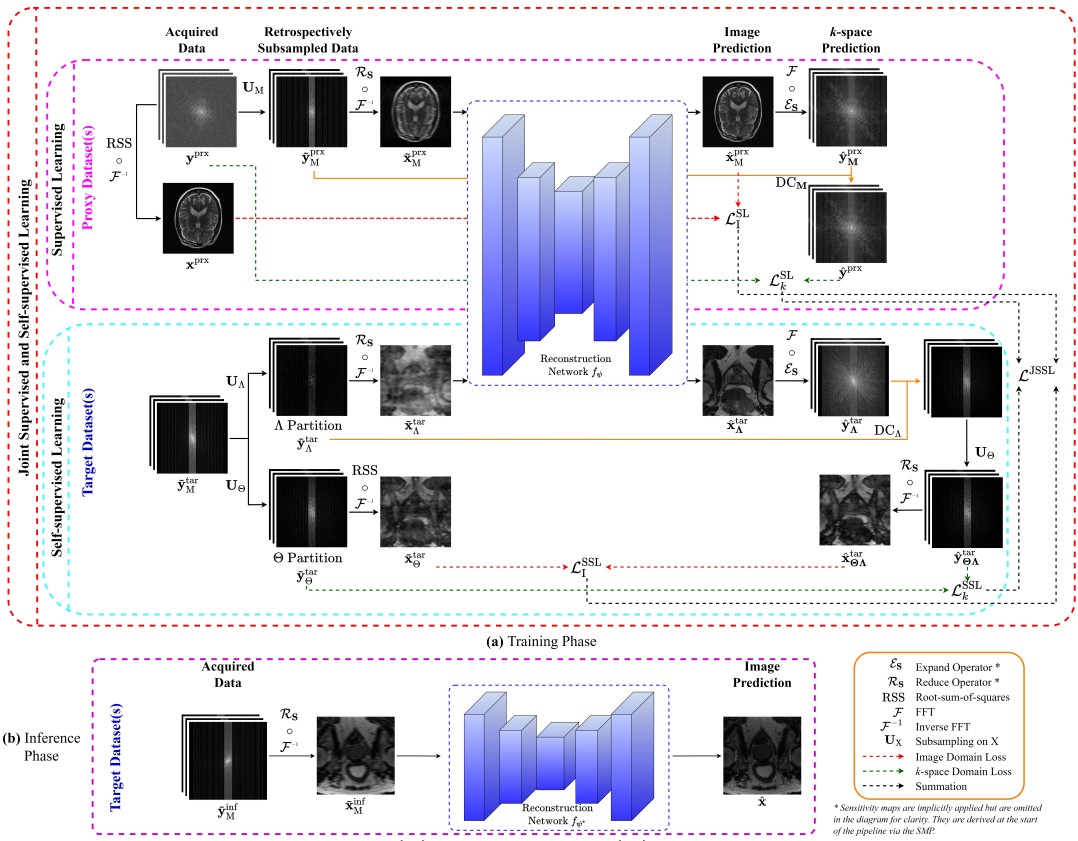

Figure 1: JSSL **(a)** training and **(b)** inference phases.

**JSSL Loss**   The JSSL loss is a fundamental component of our approach, defined as the combination of the SL and SSL losses: $\mathcal{L}_{\boldsymbol{\psi}}^{\mathrm{JSSL}} := \mathcal{L}_{\boldsymbol{\psi}}^{\mathrm{SL}} + \mathcal{L}_{\boldsymbol{\psi}}^{\mathrm{SSL}}$ and the model's parameters are updated during training such that $\boldsymbol{\psi}^* = \arg\min_{\boldsymbol{\psi}} \mathcal{L}_{\boldsymbol{\psi}}^{\mathrm{JSSL}}$.

**JSSL at Inference**   During the inference phase, assuming $\tilde{\mathbf{y}}_{\mathbf{M}}^{\mathrm{inf}}$ denotes the subsampled $k$-space data, for subsampled $k$-space data $\tilde{\mathbf{y}}_{\mathbf{M}}^{\mathrm{inf}}$ the trained JSSL reconstruction model $f_{\boldsymbol{\psi}^*}$ estimates the underlying image as follows: $\hat{\mathbf{x}} = \left| f_{\boldsymbol{\psi}^*}(\tilde{\mathbf{x}}_{\mathbf{M}}^{\mathrm{inf}}) \right|$, where $\tilde{\mathbf{x}}_{\mathbf{M}}^{\mathrm{inf}} = \mathcal{R}_{\mathbf{S}} \circ \mathcal{F}^{-1}\left(\tilde{\mathbf{y}}_{\mathbf{M}}^{\mathrm{inf}}\right)$.

## 2.5. Coil Sensitivity Prediction

An initial approximation of coil sensitivity maps (**S**) is derived from the autocalibration signal (ACS) (McKenzie et al., 2002). While SSL-based approaches use this approximation (Yaman et al., 2020; Desai et al., 2023; Zhou et al., 2022b), or employ expensive algorithms like Espirit (Uecker et al., 2013), our JSSL approach takes this initial estimation and feeds it as input to a Sensitivity Map Estimator (SME) similarly to (Millard and Chiew, 2022; Zhang et al., 2024; Hu et al., 2024). The SME aims to refine the sensitivity maps and it is trained end-to-end with the reconstruction model and is integrated in all training setups.

## 3. Experiments

### 3.1. Datasets

We utilized fully-sampled multi-coil $k$-space data from the fastMRI brain, fastMRI knee (Zbontar et al., 2019), fastMRI prostate T2 (Tibrewala et al., 2023), and CMRxRecon 2023

cardiac cine MRI (Wang et al., 2024) datasets. Their characteristics and data-splitting parameters are summarized in Tab. S1. To evaluate JSSL, we paired target and proxy datasets in two experimental sets: **(A) Target:** Prostate, **Proxy:** Brain + Knee; **(B) Target:** Cardiac, **Proxy:** Brain + Knee + Prostate. During training, target datasets were retrospectively subsampled, with fully-sampled target data reserved only for inference. Proxy datasets were also retrospectively subsampled for training, with their fully-sampled measurements used to compute the SL loss component of $\mathcal{L}_{\psi}^{\mathrm{JSSL}}$.

## 3.2. Subsampling Schemes

For our experiments, we applied random uniform subsampling to brain data and used equispaced subsampling for the knee, prostate, and cardiac data, commonly employed in the literature (Zbontar et al., 2019; Wang et al., 2024). During training, we randomly selected acceleration factors of $R = 4, 8, 12$ (only for **A**), and 16 (only for **B**). We retained 8%, 4%, 3%, and 2% of the fully-sampled ACS lines for $R = 4, 8, 12$, and 16, respectively. During inference, all these acceleration factors were tested.

**SSL Subsampling Partitioning** During the training of any SSL-based method, including JSSL, we split the subsampled data into two distinct sets, as explained in Sec. 2.3. Specifically, $\boldsymbol{\Theta}_i$ was obtained from $\mathbf{M}_i$ using a 2D Gaussian sampling approach with a standard deviation of 3.5 pixels, as it has been shown to outperform uniform partitioning (Yaman et al., 2020). Consequently, we set $\boldsymbol{\Lambda}_i = \mathbf{M}_i \setminus \boldsymbol{\Theta}_i$. Furthermore, the ratio $q_i = \frac{|\boldsymbol{\Theta}_i|}{|\mathbf{M}_i|}$ was chosen randomly between 0.3 and 0.8. Note that each $\boldsymbol{\Lambda}_i$ included a $w \times w = 4 \times 4$ window in the ACS region to enhance SME module training.

## 3.3. Implementation & Optimization

**Model Architecture** We employed vSHARP, a physics-guided deep learning approach unrolled over $T = 12$ iterations, previously used for accelerated MRI reconstruction (Yiasemis et al., 2023, 2025; Lyu et al., 2024). Each iteration's network $\{\mathcal{H}_{\boldsymbol{\theta}_t}\}_{t=0}^{T-1}$ was a U-Net (Ronneberger et al., 2015) with four scales and 32 filters at the first scale. For the data consistency step we set $T_{\mathbf{x}} = 10$. The SME module used a U-Net with 16 filters at the first scale. JSSL is model-agnostic, and we explore additional architectures in Appendix D.1.
**Optimization** We employed the Adam optimizer with $\epsilon = 10^{-8}$, $(\beta_1, \beta_2) = (0.99, 0.999)$, and an initial learning rate (lr) of 0.003. A lr scheduler reduced it by 0.8 every 150,000 iterations. Training was conducted on two A6000 RTX GPUs with a batch size of two slices per GPU using the DIRECT toolkit (Yiasemis et al., 2022a). All models were trained to convergence. Loss was computed combining image and frequency domain components motivated by prior work (Yiasemis et al., 2025):

$$\mathcal{L}_{\mathrm{I}}^{\mathrm{SL}}, \mathcal{L}_{\mathrm{I}}^{\mathrm{SSL}} := 2\left(\mathcal{L}_{\mathrm{SSIM}} + \mathcal{L}_1\right) + \mathrm{HFEN}_1 + \mathrm{HFEN}_2, \ \mathcal{L}_K^{\mathrm{SL}}, \mathcal{L}_K^{\mathrm{SSL}} := 2\left(\mathrm{NMSE} + \mathrm{NMAE}\right).$$

## 3.4. Training Setups Comparison

We conducted the following experiments: **(1)** SSL in the target domain, **(2)** SSL in both target and proxy domains (SSL ALL), **(3)** SL in the target domain, **(4)** SL in both proxy and target domains (SL ALL), **(5)** SL in proxy domains only (SL PROXY), and **(6)** JSSL.

Our primary goal was to assess JSSL against SSL approaches in scenarios where fully-sampled target data are unavailable. To verify that JSSL's performance does not simply result from using a larger dataset, we included SSL ALL, which incorporates all available data (target + proxy) under a SSL strategy. SL methods served as a reference, though their results are naturally expected to be superior when fully-sampled target data are accessible.

### 3.5. Evaluation

The performance of our experiments was evaluated on the target test sets using three metrics: SSIM, PSNR, NMSE (Yiasemis et al., 2024). Model checkpoints were selected based on validation set performance. Statistical tests assessed whether the top method in each category (SL, SSL including JSSL) significantly outperformed others. We first computed performance differences between the best and other methods within each category. The Shapiro-Wilk test checked normality; if satisfied ($p > \alpha = 0.05$), a paired t-test was used; otherwise, the Wilcoxon signed-rank test was applied. Results where the best method (**bold**) was not statistically superior ($p > \alpha = 0.05$) are marked with an asterisk ($*$).

### 3.6. Results

Table 1: Results for fastMRI prostate (target) using brain and knee (proxy) datasets.

| Setup | 2x | | | 4x | | | 8x | | | 16x | | |
|---|---|---|---|---|---|---|---|---|---|---|---|---|
| | SSIM | pSNR | NMSE | SSIM | pSNR | NMSE | SSIM | pSNR | NMSE | SSIM | pSNR | NMSE |
| SL | $\mathbf{0.974_{\pm0.010}}$ | $\mathbf{41.8_{\pm2.3}}$ | $\mathbf{0.002_{\pm0.001}}$ | $\mathbf{0.930_{\pm0.022}}$ | $\mathbf{37.5_{\pm1.8}}$ | $\mathbf{0.005_{\pm0.002}}$ | $\mathbf{0.868_{\pm0.033}}$ | $\mathbf{33.9_{\pm1.6}}$ | $\mathbf{0.011_{\pm0.003}}$ | $\mathbf{0.799_{\pm0.045}}$ | $\mathbf{31.0_{\pm1.6}}$ | $\mathbf{0.021_{\pm0.005}}$ |
| SL ALL | $0.969_{\pm0.012}$ | $41.1_{\pm2.3}$ | $0.002_{\pm0.001}$ | $0.922_{\pm0.024}$ | $36.9_{\pm1.8}$ | $0.005_{\pm0.002}$ | $0.854_{\pm0.035}$ | $33.2_{\pm1.5}$ | $0.013_{\pm0.003}$ | $0.771_{\pm0.049}$ | $30.0_{\pm1.6}$ | $0.026_{\pm0.006}$ |
| SL PROXY | $0.961_{\pm0.016}$ | $39.8_{\pm2.4}$ | $0.003_{\pm0.002}$ | $0.914_{\pm0.026}$ | $36.4_{\pm1.8}$ | $0.006_{\pm0.002}$ | $0.839_{\pm0.041}$ | $32.5_{\pm1.7}$ | $0.015_{\pm0.004}$ | $0.733_{\pm0.051}$ | $28.6_{\pm1.5}$ | $0.035_{\pm0.008}$ |
| SSL | $0.956_{\pm0.015}$ | $38.8_{\pm2.6}$ | $0.004_{\pm0.002}$ | $0.891_{\pm0.030}$ | $34.7_{\pm2.0}$ | $0.009_{\pm0.003}$ | $0.801_{\pm0.038}$ | $31.1_{\pm1.5}$ | $0.020_{\pm0.005}$ | $0.707_{\pm0.050}$ | $28.0_{\pm1.6}$ | $0.041_{\pm0.008}$ |
| SSL ALL | $0.953_{\pm0.016}$ | $38.6_{\pm2.5}$ | $0.004_{\pm0.002}$ | $0.892_{\pm0.031}$ | $34.8_{\pm2.0}$ | $0.009_{\pm0.004}$ | $0.801_{\pm0.041}$ | $31.1_{\pm1.6}$ | $0.020_{\pm0.006}$ | $0.699_{\pm0.052}$ | $27.8_{\pm1.6}$ | $0.043_{\pm0.010}$ |
| JSSL | $\mathbf{0.965_{\pm0.015}}$ | $\mathbf{39.5_{\pm2.8}}$ | $\mathbf{0.003_{\pm0.002}}$ | $\mathbf{0.918_{\pm0.026}}$ | $\mathbf{36.4_{\pm1.9}}$ | $\mathbf{0.006_{\pm0.002}}$ | $\mathbf{0.842_{\pm0.038}}$ | $\mathbf{32.5_{\pm1.6}}$ | $\mathbf{0.015_{\pm0.004}}$ | $\mathbf{0.752_{\pm0.053}}$ | $\mathbf{29.3_{\pm1.6}}$ | $\mathbf{0.030_{\pm0.007}}$ |

Table 2: Results for cardiac (target) using brain, knee and prostate (proxy) datasets.

| Setup | 2x | | | 4x | | | 8x | | | 12x | | |
|---|---|---|---|---|---|---|---|---|---|---|---|---|
| | SSIM | pSNR | NMSE | SSIM | pSNR | NMSE | SSIM | pSNR | NMSE | SSIM | pSNR | NMSE |
| SL | $\mathbf{0.991_{\pm0.003}}$ | $\mathbf{48.1_{\pm2.5}}$ | $\mathbf{0.002_{\pm0.003}}$ | $\mathbf{0.984_{\pm0.005}}$ | $\mathbf{45.7_{\pm2.0}}$ | $\mathbf{0.006_{\pm0.002}}$ | $\mathbf{0.965_{\pm0.011}}$ | $\mathbf{40.6_{\pm2.2}}$ | $\mathbf{0.018_{\pm0.007}}$ | $\mathbf{0.946_{\pm0.018}}$ | $\mathbf{37.8_{\pm2.3}}$ | $\mathbf{0.035_{\pm0.015}}$ |
| SL ALL | $0.987_{\pm0.004}$ | $46.5_{\pm2.6}$ | $0.005_{\pm0.004}$ | $0.979_{\pm0.006}$ | $44.5_{\pm1.9}$ | $0.007_{\pm0.003}$ | $0.956_{\pm0.012}$ | $39.4_{\pm1.9}$ | $0.024_{\pm0.008}$ | $0.932_{\pm0.019}$ | $36.5_{\pm2.0}$ | $0.047_{\pm0.016}$ |
| SL PROXY | $0.875_{\pm0.037}$ | $39.8_{\pm2.0}$ | $0.022_{\pm0.009}$ | $0.880_{\pm0.035}$ | $37.6_{\pm2.0}$ | $0.036_{\pm0.012}$ | $0.848_{\pm0.034}$ | $33.1_{\pm1.7}$ | $0.099_{\pm0.027}$ | $0.810_{\pm0.041}$ | $30.0_{\pm2.2}$ | $0.211_{\pm0.079}$ |
| SSL | $0.944_{\pm0.017}$ | $41.2_{\pm2.1}$ | $0.016_{\pm0.007}$ | $0.902_{\pm0.020}$ | $36.2_{\pm2.0}$ | $0.049_{\pm0.014}$ | $0.854_{\pm0.025}$ | $33.2_{\pm1.7}$ | $0.097_{\pm0.020}$ | $0.817_{\pm0.032}$ | $\mathbf{31.2_{\pm1.9}}$ | $\mathbf{0.153_{\pm0.038}}$ |
| SSL ALL | $0.974_{\pm0.006}$ | $44.0_{\pm1.9}$ | $0.009_{\pm0.005}$ | $0.929_{\pm0.016}$ | $37.9_{\pm1.9}$ | $0.033_{\pm0.011}$ | $0.862_{\pm0.026}$ | $33.0_{\pm1.7}$ | $0.102_{\pm0.026}$ | $0.814_{\pm0.034}$ | $30.3_{\pm2.0}$ | $0.191_{\pm0.059}$ |
| JSSL | $\mathbf{0.975_{\pm0.007}}$ | $\mathbf{45.5_{\pm2.0}}$ | $\mathbf{0.006_{\pm0.004}}$ | $\mathbf{0.944_{\pm0.013}}$ | $\mathbf{39.2_{\pm2.0}}$ | $\mathbf{0.025_{\pm0.010}}$ | $\mathbf{0.893_{\pm0.022}}$ | $\mathbf{34.3_{\pm1.8}}$ | $\mathbf{0.077_{\pm0.023}}$ | $\mathbf{0.848_{\pm0.032}}$ | $31.1_{\pm2.1}{}^{*}$ | $0.161_{\pm0.059}{}^{*}$ |

The quantitative results of our comparative studies are summarized in Tables 1 and 2, which detail metric averages and statistical significance. As expected, supervised methods consistently produced the best reconstruction results across both experimental setups.

From Tab. 1, it is evident that in experiment set **A** (prostate as target), JSSL demonstrated superior reconstruction performance across all acceleration factors and metrics compared to both SSL and SSL utilizing all data (SSL ALL). Notably, JSSL approached the performance of supervised methods (SL, SL ALL), particularly at $R = 2, 4, 8$. The use of proxy datasets in SSL settings (SSL ALL) did not enhance performance over SSL alone. Similarly, supervised training on all data (SL ALL) offered no significant advantage over SL alone. In SL PROXY, where training relied solely on proxy datasets, out-of-distribution inference on the prostate dataset resulted in better reconstruction quality tha SSL. However, JSSL outperformed SL PROXY in SSIM across all acceleration factors and matched or exceeded pSNR and NMSE, except at $R = 2$, where SL PROXY showed a slight edge.

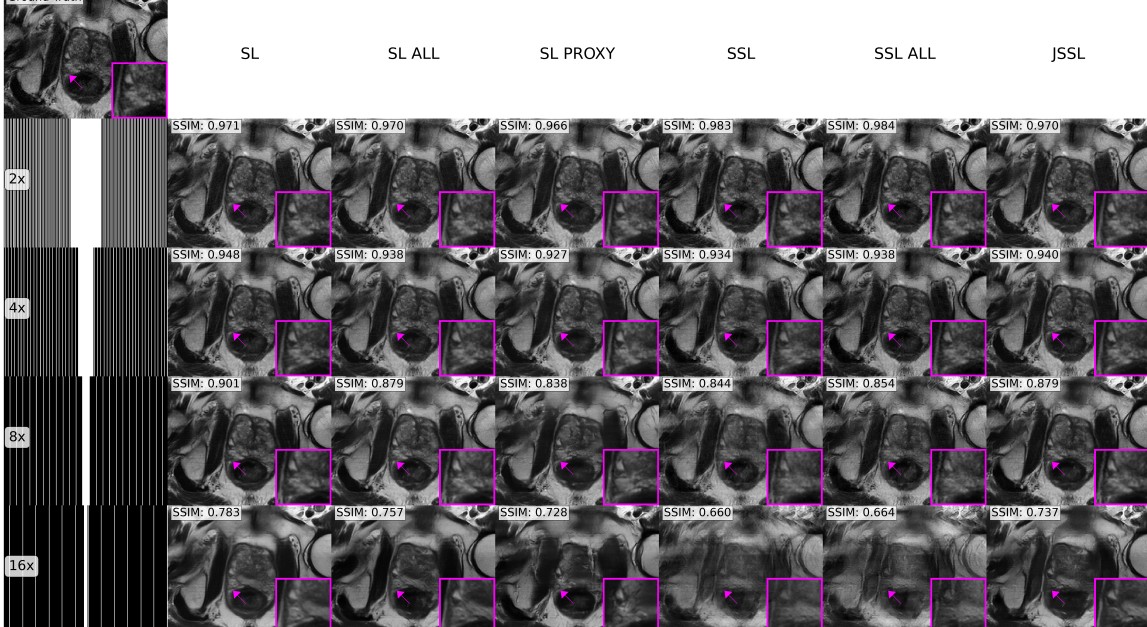

Figure 2: Example reconstructions from experiment set **A** (prostate MRI) across different training setups. The zoomed-in region highlights a clinically significant PIRADS 4 lesion (indicated by an arrow and bounding box). JSSL preserves lesion visibility even at high accelerations, whereas SSL reconstructions exhibit oversmoothing and blurring artifacts.

In experiment set **B** (cardiac as target), similar patterns were observed. JSSL consistently outperformed other SSL methods, except at $R = 12$, where SSL achieved slightly better (but non-significant) pSNR and NMSE, as shown in Tab. 2. Unlike in **A**, SSL ALL showed performance improvements over SSL for cardiac data. SL PROXY, however, performed worse than all other methods.

For qualitative analysis, Figures 2, 3, S2, and S3 display sample reconstructions. At lower accelerations ($R = 2, 4$), all methods accurately reconstructed prostate data. At higher accelerations, supervised, SL PROXY, and JSSL setups exhibited fewer artifacts compared to SSL and SSL ALL. A similar trend was observed for cardiac data, where SSL-based reconstructions were visually weaker, particularly at high accelerations ($R = 8, 12$), yielding highly aliased images. Consistent with the quantitative results, out-of-distribution inference (SL PROXY) reconstructions exhibited noticeable artifacts.

## 4. Discussion and Conclusion

This study introduces Joint Supervised and Self-supervised Learning, a novel training framework aimed at improving MRI reconstruction quality when fully-sampled $k$-space data are unavailable for the target domain. By integrating SL on fully-sampled proxy datasets with SSL on subsampled target datasets, JSSL offers a practical alternative to SSL methods, achieving superior reconstruction quality when acquiring fully-sampled data is infeasible.

Our results demonstrate that JSSL consistently yields higher reconstruction quality across various accelerations, even when proxy datasets differ anatomically from the target dataset. Beyond quantitative improvements, JSSL also enhances the clinical interpretabil-

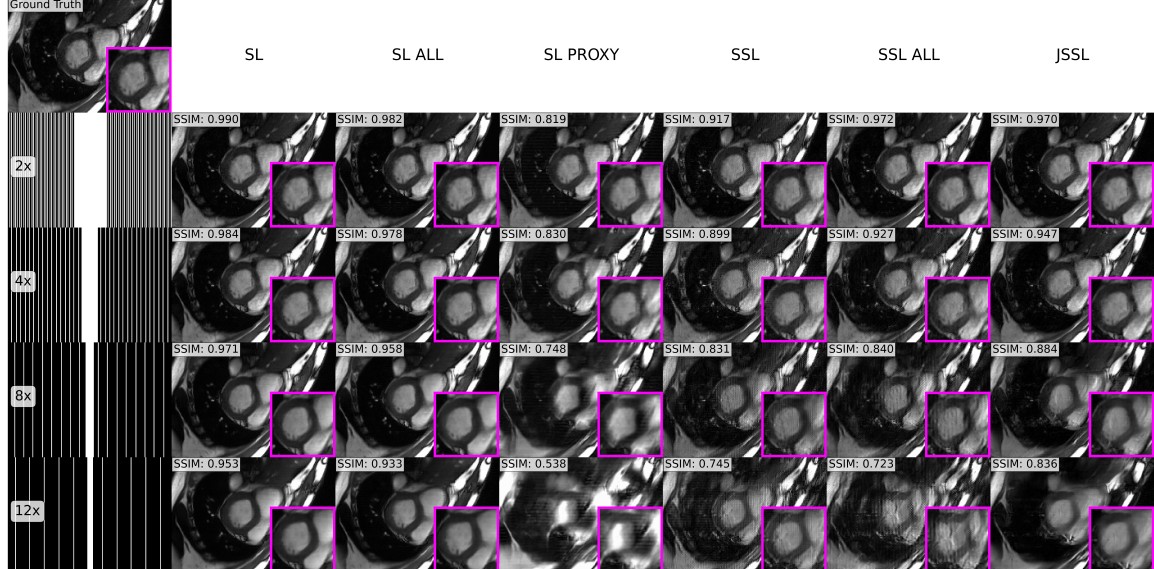

Figure 3: Example reconstructions from experiment set **B** (cardiac MRI) across different training setups. The zoomed-in region focuses on the heart, showing that JSSL maintains sharper anatomical boundaries and clearer phase transitions, while SSL reconstructions suffer from blurring and structural loss.

ity of reconstructed images. As observed in Figures 2 and 3, JSSL better preserves key anatomical structures across different accelerations compared to SSL. In the prostate samples, lesion visibility is maintained even at high acceleration rates, crucial for detecting clinically significant cancer. Similarly, in cardiac MRI, JSSL reconstructions exhibit clearer heart boundaries and cardiac phase transitions, reducing artifacts that could impair clinical assessment. Additionally, JSSL achieves consistent improvements across different model architectures, showcasing its robustness and independence from specific architectural choices.

The effectiveness of JSSL is influenced by the choice and similarity of proxy datasets. For instance, in our experiments SL PROXY struggled when proxies were highly dissimilar from the target domain. Moreover, incorporating proxy datasets may introduce biases that could impact model performance. Additionally, the partitioning strategy for self-supervised learning, the choice of loss functions, and their weighting in JSSL training may further affect results. Lastly, our comparisons are limited to SSDU as a representative SSL method, given that most self-supervised approaches are derivatives of SSDU. Extended discussion of limitations in Appendix F. Ultimately, JSSL aims to enhance SSL performance and *not* to compete with SL in cases where fully-sampled ground truth data for the target domain are available, as SL remains the optimal choice under such conditions. Based on our findings, we propose the following "rule-of-thumb" training guidelines:

(1) Use SL when fully-sampled ground truth data are available for the target dataset.
(2) When only subsampled target data are present, and ground truth data are accessible from proxy datasets (e.g., fastMRI or CMRxRecon), adopt the JSSL approach.
(3) If only proxy ground truth data exist, supervised training in proxy domains can be effective, particularly when proxies are anatomically similar to the target domain.
(4) In scenarios with only subsampled target data proceed with SSL.

## Acknowledgments

This work was supported by institutional grants of the Dutch Cancer Society and of the Dutch Ministry of Health, Welfare and Sport. The authors would like to acknowledge the Research High Performance Computing (RHPC) facility of the Netherlands Cancer Institute (NKI).

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

## Appendix A. Related Work

In the realm of self-supervised learning-based MRI reconstruction, among the first works introduced was SSDU (Self-supervised learning via data undersampling) (Yaman et al., 2020). SSDU, inspired by SSL concepts from deep learning, particularly Noise2Self (Batson and Royer, 2019), proposed training a reconstruction model (ResNet CNN with conjugate gradient formulation) by partitioning the undersampled data into two subsets. One subset served as input, and the other as the target, with the loss estimated in the $k$-space domain.

An extension of SSDU was proposed in a parallel network framework (Hu et al., 2021), where two networks were trained on each partition of the subsampled $k$-space data. A consistency loss minimized the discrepancy between the two networks' outputs, allowing either network to be used during inference since both networks were trained to produce consistent results.

Further building on SSDU, (Millard and Chiew, 2022) introduced a Noisier2Noise framework, where a second subsampling mask was applied to the already subsampled $k$-space data. The employed network, E2EVarNet (Sriram et al., 2020), was trained to recover singly subsampled data from the doubly subsampled version, showing that SSDU is a special case of this broader method. Furthermore, (Millard and Chiew, 2022) provided theoretical justifications for SSDU.

In the realm of diffusion-based MRI reconstruction, a fully-sampled-data-free score-based diffusion model was proposed in (Cui et al., 2022), where the model learned the prior of fully-sampled images from subsampled data in a self-supervised manner. Another diffusion-based approach, SSDiffRecon (Korkmaz et al., 2023), integrated cross-attention transformers with data-consistency blocks in an unrolled architecture. However, diffusion-based methods are outside the scope of our work.

Following the SSDU subsampled data splitting, in (Yan et al., 2023) the authors present DC-SiamNet, which employs two branches with shared weights in a Siamese architecture. Each branch reconstructs from a partition of the $k$-space data, and the training is guided by a dual-domain loss that includes image and frequency domain consistency which ensure reconstructed images/$k$-spaces are consistent across partitions, along with contrastive loss in the latent space.

A more recent work extended SSDU by introducing SPICER, which includes coil sensitivity estimation based on autocalibration signal (ACS) data and utilizes U-Net-based models for both sensitivity estimation and reconstruction (Hu et al., 2024). Similar sensitivity estimation was also employed in (Millard and Chiew, 2022) within the E2EVarNet framework.

Finally, SSDU has also been applied to reconstruct non-Cartesian MRI data, with the subsampled $k$-space split into disjoint parts (Zhou et al., 2022b). In this approach, a variational network is trained using a dual-domain loss similar to (Yan et al., 2023): frequency consistency ensures that reconstructed $k$-spaces from each partition match the input data, while image consistency ensures that the reconstructed images are consistent across partitions. Additionally, loss is computed by comparing the reconstructed $k$-spaces and images from each partition with those generated when subsampled data is used as input.

Most self-supervised MRI reconstruction methods can be seen as derivatives or extensions of SSDU, with partitioning of undersampled data into disjoint subsets as the funda-

mental idea. This partitioning approach underpins the SSL component of our method, and without loss of generality, SSDU can be considered a representative method in this domain. While recent techniques have incorporated different architectures or loss functions, they largely build upon this core strategy.

Our proposed method, Joint Supervised and Self-Supervised Learning, draws inspiration from these aforementioned approaches. Like most SSL-based methods, it seeks to overcome the challenge of training without fully-sampled $k$-space data for the target organ domain. However, JSSL extends the applicability of such techniques by leveraging fully-sampled data from proxy datasets while incorporating subsampled data from the target domain. This enables joint training through both supervised and self-supervised learning, providing a practical solution for scenarios where ground truth fully-sampled data is inaccessible, yet allowing for improved reconstruction performance through the combination of proxy and target datasets.

In the broader context of combining supervised and self-supervised learning, Noise2Recon (Desai et al., 2023) extended SSDU by leveraging both fully-sampled and subsampled data within a single organ domain for reconstruction and denoising, using the E2EVarNet model (Sriram et al., 2020). However, this method's dependency on fully-sampled data restricts its applicability in scenarios where such data is unavailable.

Another recent approach utilized paired fully-sampled and subsampled data from different modalities for reconstruction of the target modality (Zhou et al., 2022a). While SSL was employed for training, this method still relied on fully-sampled data during both training and inference, which contrasts with our approach that focuses on cases where fully-sampled data is unavailable for the target domain.

Lastly, test-time training (Darestani et al., 2022) is a recent method proposed to handle domain shifts in MRI reconstruction. By re-training models at inference times using a SSL data-consistency loss, it aims to adjust to shifts in data distribution between training and testing, such as moving from one scanner to another. However, this technique operates at inference time, which limits its utility in real-time imaging applications.

## Appendix B. JSSL Theoretical Motivation

The core concept behind JSSL is to leverage both supervised and self-supervised learning to enhance MRI reconstruction of a target dataset, even when the parameters optimized on supervised proxy tasks may not be the most optimal. We hypothesize that introducing a supervised proxy task serves as a form of regularization, reducing the variance of our estimators due to the proxy supervised training on a 'less noisy' task. We illustrate this intuition with two simplified examples in Proposition 1 (estimating means of distributions) and Proposition 2 (linear regression), where we assume two distributions - one that we wish to estimate, but we cannot obtain sufficient samples from, and a proxy distribution that is directly accessible. We demonstrate that drawing samples from both distributions (or using only the proxy distribution) can reduce our estimator's variance and risk.

**Proposition 1** *Consider two distributions $p_i$, $i = 1, 2$ with means and variances $\mu_i, \sigma_i$, $i = 1, 2$, with unknown $\mu_1$, and $\mu_1 \neq \mu_2$. Then if $(\mu_1 - \mu_2)^2 < c\frac{\sigma_1^2}{N}$ for some $c \in (0, 1)$ and*

$N \in \mathbb{Z}^+$, then $\tilde{x} = \frac{1}{N+K} \sum_{i=1}^{N+K} x_i$ is a lower-variance estimator of $\mu_1$ compared to $\overline{x} = \frac{1}{N} \sum_{i=1}^{N} x_i$, where $\left\{ x^{(i)} \sim p_1 \right\}_{i=1}^{N}$ and $\left\{ x^{(N+i)} \sim p_2 \right\}_{i=1}^{K}$ for a choice of a large $K \in \mathbb{Z}^+$.

**Proof** Assume a mixture distribution:

$$p_\pi(x) = \pi \mathcal{N}(x|\mu_1, \sigma_1^2) + (1 - \pi)\mathcal{N}(x|\mu_2, \sigma_2^2).$$

It is then straightforward to compute:

$$\mathbb{E}\left[p_\pi\right] = \pi\mu_1 + (1 - \pi)\mu_2$$

and,

$$\mathbb{V}\left[p_\pi\right] = \pi\sigma_1^2 + (1 - \pi)\sigma_2^2 + \pi(1 - \pi)(\mu_2 - \mu_1)^2.$$

Drawing $\left\{ x^{(i)} \sim p_1 \right\}_{i=1}^{N}$ and $\left\{ x^{(N+i)} \sim p_2 \right\}_{i=1}^{K}$, is approximately equivalent to drawing $N + K$ samples from the mixture $p_\pi$ with $\pi = \frac{N}{N+K}$. Using bias-variance decomposition, we can compute the expected mean squared errors for the two estimators:

$$\mathbb{E}\left[(\overline{x} - \mu_1)^2\right] = \frac{\sigma_1^2}{N},$$

and,

$$\mathbb{E}\left[(\tilde{x} - \mu_1)^2\right] = (1 - \pi)^2(\mu_1 - \mu_2)^2 + \frac{\pi\sigma_1^2 + (1 - \pi)\sigma_2^2 + \pi(1 - \pi)(\mu_1 - \mu_2)^2}{N + K}.$$

If $(\mu_1 - \mu_2)^2 < c\frac{\sigma_1^2}{N}$ for some $c \in (0, 1)$, then taking the limit $K \to \infty$ and thus $\pi \to 0$, we observe that

$$\mathbb{E}\left[(\tilde{x} - \mu_1)^2\right] \to (\mu_1 - \mu_2)^2 < c\frac{\sigma_1^2}{N} < \frac{\sigma_1^2}{N} = \mathbb{E}\left[(\overline{x} - \mu_1)^2\right].$$

■

**Proposition 2** *Let $\boldsymbol{x} \sim \mathcal{N}(\boldsymbol{0}, \sigma^2 \boldsymbol{I}_p)$ be $\mathbb{R}^p$-valued isotropic Gaussian random vector and $y, \tilde{y}$ be random variables with $p(y|\boldsymbol{x}) = \mathcal{N}(y|\boldsymbol{w}^T \boldsymbol{x}, \varepsilon^2)$ and $p(\tilde{y}|\boldsymbol{x}) = \mathcal{N}(\tilde{y}|\tilde{\boldsymbol{w}}^T \boldsymbol{x}, \tilde{\varepsilon}^2)$ for some $\boldsymbol{w}, \tilde{\boldsymbol{w}} \in \mathbb{R}^p$. Let $\mathcal{T} = \{(\boldsymbol{x}_1, \tilde{y}_1), \dots, (\boldsymbol{x}_K, \tilde{y}_K)\}$ be a training data set with $K > p$ and consider a maximum likelihood estimator $\widehat{y}(\boldsymbol{x}; \mathcal{T})$ for $y$ given $\boldsymbol{x}$, computed using $\mathcal{T}$. Then the following holds:*

1. $\mathrm{Bias}_{\mathcal{T}}[\widehat{y}(\boldsymbol{x}; \mathcal{T})] = (\tilde{\boldsymbol{w}}^T - \boldsymbol{w}^T)\boldsymbol{x}$.
2. $\mathrm{Var}_{\mathcal{T}}[\widehat{y}(\boldsymbol{x}; \mathcal{T})] = \frac{\tilde{\varepsilon}^2}{\sigma^2 K}\|\boldsymbol{x}\|_2^2$.
3. $\mathbb{E}_{(\boldsymbol{x}, y)}[\widehat{y}(\boldsymbol{x}; \mathcal{T}) - y]^2 \leq p\sigma^2\|\tilde{\boldsymbol{w}} - \boldsymbol{w}\|_2^2 + \frac{p\tilde{\varepsilon}^2}{K} + \varepsilon^2$

**Proof** Let $\tilde{\boldsymbol{w}}_{\mathrm{MLE}} = (\boldsymbol{X}^T \boldsymbol{X})^{-1} \boldsymbol{X}^T \tilde{\boldsymbol{y}}$ be the MLE estimator for $\tilde{\boldsymbol{w}}$, where the $K$ rows of $\boldsymbol{X} \in \mathbb{R}^{K \times p}$ are given by $\boldsymbol{x}_1^T, \dots, \boldsymbol{x}_K^T$ and the vector $\tilde{\boldsymbol{y}}$ is defined as $\tilde{\boldsymbol{y}} := (\tilde{y}_1, \dots, \tilde{y}_K) \in \mathbb{R}^K$. Since $K > p$, matrix $\boldsymbol{X}$ has full column rank almost surely and thus $\boldsymbol{X}^T \boldsymbol{X}$ is almost surely invertible. Observe that

$$\mathbb{E}_{\mathcal{T}}[\tilde{\boldsymbol{w}}_{\mathrm{MLE}}^T] = \mathbb{E}_{\mathcal{T}}[(\tilde{\boldsymbol{\varepsilon}}^T + \tilde{\boldsymbol{w}}^T \boldsymbol{X}^T)\boldsymbol{X}(\boldsymbol{X}^T \boldsymbol{X})^{-1}] = \tilde{\boldsymbol{w}}^T,$$

since $\tilde{\boldsymbol{\varepsilon}} := \tilde{\boldsymbol{y}} - \boldsymbol{X}\tilde{\boldsymbol{w}}$ has zero mean, is independent from $\boldsymbol{x}_i$'s and the expectation $\mathbb{E}_{\mathcal{T}}[\cdot]$ can be rewritten as $\mathbb{E}_{\boldsymbol{x}_1, \dots, \boldsymbol{x}_K}[\mathbb{E}_{\tilde{\boldsymbol{\varepsilon}}}[\cdot]]$. By definition of estimator bias,

$$\mathrm{Bias}_{\mathcal{T}}[\widehat{y}(\boldsymbol{x}; \mathcal{T})] = \mathbb{E}_{\mathcal{T}}[\widehat{y}(\boldsymbol{x}; \mathcal{T})] - \mathbb{E}_{y|\boldsymbol{x}}y = \mathbb{E}_{\mathcal{T}}[\tilde{\boldsymbol{w}}_{\mathrm{MLE}}^T]\boldsymbol{x} - \boldsymbol{w}^T \boldsymbol{x} = (\tilde{\boldsymbol{w}}^T - \boldsymbol{w}^T)\boldsymbol{x}.$$

Next,

$$\mathrm{Var}_{\mathcal{T}}[\widehat{y}(\boldsymbol{x}; \mathcal{T})] = \mathbb{E}_{\mathcal{T}}[\mathbb{E}_{\mathcal{T}}[\widehat{y}(\boldsymbol{x}; \mathcal{T})] - \widehat{y}(\boldsymbol{x}; \mathcal{T})]^2 =$$
$$= \mathbb{E}_{\mathcal{T}}[\tilde{\boldsymbol{w}}^T \boldsymbol{x} - (\tilde{\boldsymbol{\varepsilon}}^T + \tilde{\boldsymbol{w}}^T \boldsymbol{X}^T)\boldsymbol{X}(\boldsymbol{X}^T \boldsymbol{X})^{-1}\boldsymbol{x}]^2 = \mathbb{E}_{\mathcal{T}}[\tilde{\boldsymbol{\varepsilon}}^T \boldsymbol{X}(\boldsymbol{X}^T \boldsymbol{X})^{-1}\boldsymbol{x}]^2.$$

The scalar $(\tilde{\boldsymbol{\varepsilon}}^T \boldsymbol{X}(\boldsymbol{X}^T \boldsymbol{X})^{-1}\boldsymbol{x})^2$ can be equivalently written as

$$(\tilde{\boldsymbol{\varepsilon}}^T \boldsymbol{X}(\boldsymbol{X}^T \boldsymbol{X})^{-1}\boldsymbol{x})^T(\tilde{\boldsymbol{\varepsilon}}^T \boldsymbol{X}(\boldsymbol{X}^T \boldsymbol{X})^{-1}\boldsymbol{x}) = \boldsymbol{x}^T(\boldsymbol{X}^T \boldsymbol{X})^{-1}\boldsymbol{X}^T \tilde{\boldsymbol{\varepsilon}}\tilde{\boldsymbol{\varepsilon}}^T \boldsymbol{X}(\boldsymbol{X}^T \boldsymbol{X})^{-1}\boldsymbol{x}.$$

Using that $\mathbb{E}_{\mathcal{T}}[\cdot] = \mathbb{E}_{\boldsymbol{x}_1, \dots, \boldsymbol{x}_k}[\mathbb{E}_{\tilde{\boldsymbol{\varepsilon}}}[\cdot]]$, we deduce that

$$\mathbb{E}_{\mathcal{T}}[\tilde{\boldsymbol{\varepsilon}}^T \boldsymbol{X}(\boldsymbol{X}^T \boldsymbol{X})^{-1}\boldsymbol{x}]^2 = \mathbb{E}_{\boldsymbol{x}_1, \dots, \boldsymbol{x}_K}[\boldsymbol{x}^T(\boldsymbol{X}^T \boldsymbol{X})^{-1}\boldsymbol{X}^T \mathbb{E}_{\tilde{\boldsymbol{\varepsilon}}}[\tilde{\boldsymbol{\varepsilon}}\tilde{\boldsymbol{\varepsilon}}^T]\boldsymbol{X}(\boldsymbol{X}^T \boldsymbol{X})^{-1}\boldsymbol{x}] =$$
$$= \tilde{\varepsilon}^2 \mathbb{E}_{\boldsymbol{x}_1, \dots, \boldsymbol{x}_K}[\boldsymbol{x}^T(\boldsymbol{X}^T \boldsymbol{X})^{-1}\boldsymbol{x}] = \tilde{\varepsilon}^2 \mathbb{E}_{\boldsymbol{x}_1, \dots, \boldsymbol{x}_K}[\mathrm{tr}(\boldsymbol{x}^T(\boldsymbol{X}^T \boldsymbol{X})^{-1}\boldsymbol{x})] =$$
$$= \tilde{\varepsilon}^2 \mathbb{E}_{\boldsymbol{x}_1, \dots, \boldsymbol{x}_K}[\mathrm{tr}(\boldsymbol{x}\boldsymbol{x}^T(\boldsymbol{X}^T \boldsymbol{X})^{-1})] = \tilde{\varepsilon}^2 \mathrm{tr}(\boldsymbol{x}\boldsymbol{x}^T \mathbb{E}_{\boldsymbol{x}_1, \dots, \boldsymbol{x}_K}[(\boldsymbol{X}^T \boldsymbol{X})^{-1}]),$$

where we use cyclic property of the trace and the fact that $z = \mathrm{tr}(z)$ for a scalar $z$. To compute $\mathbb{E}_{\boldsymbol{x}_1, \dots, \boldsymbol{x}_K}[(\boldsymbol{X}^T \boldsymbol{X})^{-1}]$, we note that, by definition, $\boldsymbol{X}^T \boldsymbol{X}$ follows Wishart distribution $\mathcal{W}_p(\sigma^2 \boldsymbol{I}_p, K)$ with $K$ degrees of freedom and thus $(\boldsymbol{X}^T \boldsymbol{X})^{-1}$ follows inverse Wishart distribution $\mathcal{W}_p^{-1}(\sigma^{-2} \boldsymbol{I}_p, K + p + 1)$, whose mean equals $\frac{\boldsymbol{I}_p}{\sigma^2 K}$. Combining this with the previous results, we conclude

$$\mathrm{Var}_{\mathcal{T}}[\widehat{y}(\boldsymbol{x}; \mathcal{T})] = \frac{\tilde{\varepsilon}^2}{\sigma^2 K}\mathrm{tr}(\boldsymbol{x}\boldsymbol{x}^T) = \frac{\tilde{\varepsilon}^2}{\sigma^2 K}\|\boldsymbol{x}\|_2^2.$$

The final estimate follows from the first two identities and the bias-variance decomposition.
∎

Propositions 1 and 2 imply that leveraging a large number of samples from the proxy distribution ($K \to \infty$) can lead to a significant reduction in the variance of estimators trained under both supervised and self-supervised learning paradigms. Moreover, it highlights how the introduction of bias through supervised learning can be a strategic trade-off to lower variance. Additionally, Proposition 2 sheds light on how the risk associated with our estimator can be influenced by the degree of similarity between the target and proxy distributions.

## Appendix C. Experiments

### C.1. Datasets Information

In our experiments we utilize the fastMRI Knee (Zbontar et al., 2019), fastMRI Brain (Zbontar et al., 2019), fastMRI Prostate (Lyu et al., 2024; Wang et al., 2024), and CM-RxRecon Cine (Tibrewala et al., 2023) datasets. The characteristics and data splits are shown below in Table S1.

Table S1: Dataset characteristics and splits.

| Dataset | | fastMRI Knee | fastMRI Brain | fastMRI Prostate | CMRxRecon Cine |
|---|---|---|---|---|---|
| Field Strength | | 1.5 T, 3.0 T | 1.5 T, 3.0 T | 3.0 T | 3.0 T |
| Sequence | | Proton Density with and without fat suppression | T1-w pre and post contrast, T2-w, FLAIR | T2-w | Cine |
| Subjects | | Healthy or Abnormality present | Healthy or Pathology present | Cancer Patients | Healthy |
| Acquisition | | Cartesian | Cartesian | Cartesian | Cartesian |
| Fully Sampled or Subsampled | | Fully Sampled | Fully Sampled | Three averages (2x) / GRAPPA reconstructed | One average (3x) / GRAPPA reconstructed |
| No. Coils | | 15 | 2-24 | 10-30 | 10 |
| No. Volumes Used | | 973 | 2,991 | 312 | 473 |
| No. Slices Used | | 34,742 | 47,426 | 9,508 | 3,185 |
| Split Size (No. Volumes/ No. Slices) | Training | 973 / 34,742 | 2,991 / 47,426 | 218 / 6,647 | 203 / 1,364 |
| | Validation | - | - | 48 / 1,462 | 111 / 731 |
| | Test | - | - | 46 / 1,399 | 159 / 1,090 |

In the comparative experiments outlined in Section 3.4, we addressed the imbalance between proxy datasets (brain and knee in experiment set **A**; brain, knee, and prostate in experiment set **B**) and target datasets (prostate in experiment set **A**; cardiac cine in experiment set **B**) by oversampling the proxy data. Unless specified otherwise, this was achieved by duplicating each proxy dataset sample to ensure consistency across experiments.

### C.2. SSL Subsampling Partitioning

Let $\mathbf{M}_i$ denote the sampling set. Here we describe $\mathbf{M}_i$ as a sampling mask in the form of a squared array of size $n = n_x \times n_y$ such that:

$$(\mathbf{M}_i)_{kj} = \begin{cases} 1, & \text{if } (k,j) \text{ is sampled} \\ 0, & \text{if } (k,j) \text{ is not sampled.} \end{cases}$$

The set $\mathbf{\Theta}_i$ is obtained by selecting elements from $\mathbf{M}_i$ using a variable density 2D Gaussian scheme with a standard deviation of $\sigma$ pixels and mean vector as the center of the sampling set $\mathbf{M}_i$, up to the number of elements determined by a ratio $q_i$, determined such that $q_i = \frac{|\mathbf{\Theta}_i|}{|\mathbf{M}_i|}$, where $|\cdot|$ here denotes the cardinality. Mathematically, the selection process for $\mathbf{\Theta}_i$ from $\mathbf{M}_i$ can be described by the following algorithm:

**Data:** Square array $\mathbf{M}_i$ of size $n_x \times n_y$, ratio $0 < q_i < 1$, standard deviation $\sigma$
**Result:** Set $\mathbf{\Theta}_i$
Initialize $\mathbf{\Theta}_i$ as an array of zeros of the same size as $\mathbf{M}_i$ **while** $\frac{|\mathbf{\Theta}_i|}{|\mathbf{M}_i|} < q_i$ **do**

    Generate $(k, j)$ from $\mathcal{N}\left([\frac{n_x}{2}, \frac{n_y}{2}], \sigma^2 \mathbf{I}_2\right)$ **if** $(\mathbf{\Theta}_i)_{kj} == 0$ **then**
    |   $(\mathbf{\Theta}_i)_{kj} \leftarrow 1$
    **end**

**end**

  **Algorithm 1:** Generation of $\mathbf{\Theta}_i$ using Gaussian Sampling

Subsequently, to partition $\mathbf{M}_i$, we set $\mathbf{\Lambda}_i = \mathbf{M}_i \smallsetminus \mathbf{\Theta}_i$. Note that by selecting $q_i = 0$ then $\mathbf{\Theta}_i = \emptyset$, and for $q_i = 1$ then $\mathbf{\Theta}_i = \mathbf{M}_i$.

For our comparison study in Section 3.6 of the main paper for SSL and JSSL experiments we randomly selected the ratio $q_i$ between 0.3, 0.4, 0.5, 0.6, 0.7 and 0.8. For our alternative configurations study in Appendix D.1, we employed an identical partitioning ratio selection except for the case of a fixed ratio of $q_i = 0.5$. In all our JSSL and SSL experiments we used $\sigma = 3.5$.

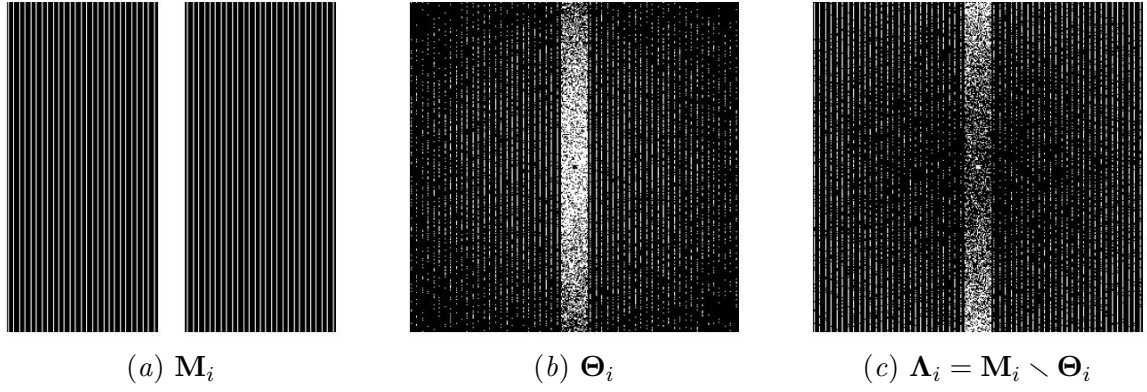

$(a)\ \mathbf{M}_i$              $(b)\ \mathbf{\Theta}_i$              $(c)\ \mathbf{\Lambda}_i = \mathbf{M}_i \smallsetminus \mathbf{\Theta}_i$

Figure S1: Example of SSL subsampling partitioning with a ratio $q = \frac{1}{2}$ and $w = 4$.

### C.3. Choice of Loss Functions

Following, we provide the mathematical definitions of each component of the loss function described in Section 3.3:

- Image Domain Loss Functions
    - Structural Similarity Index Measure (SSIM) Loss

$$\mathcal{L}_{\text{SSIM}} := 1 - \text{SSIM}, \quad \text{SSIM}(\mathbf{a}, \mathbf{b}) = \frac{1}{N} \sum_{i=1}^{N} \frac{(2\mu_{\mathbf{a}_i}\mu_{\mathbf{b}_i} + \gamma_1)(2\sigma_{\mathbf{a}_i\mathbf{b}_i} + \gamma_2)}{(\mu_{\mathbf{a}_i}^2 + \mu_{\mathbf{b}_i}^2 + \gamma_1)(\sigma_{\mathbf{a}_i}^2 + \sigma_{\mathbf{b}_i}^2 + \gamma_2)}, \tag{11}$$

where $\mathbf{a}_i, \mathbf{b}_i, i = 1, ..., N$ represent $7 \times 7$ square windows of $\mathbf{a}, \mathbf{b}$, respectively, and $\gamma_1 = 0.01$, $\gamma_1 = 0.03$. Additionally, $\mu_{\mathbf{a}_i}$, $\mu_{\mathbf{b}_i}$ denote the means of each window, $\sigma_{\mathbf{a}_i}$ and $\sigma_{\mathbf{b}_i}$ represent the corresponding standard deviations. Lastly, $\sigma_{\mathbf{a}_i\mathbf{b}_i}$ represents the covariance between $\mathbf{a}_i$ and $\mathbf{b}_i$.

– High Frequency Error Norm (HFEN)

$$\mathcal{L}_{\text{HFEN}_k} := \text{HFEN}_k, \quad \text{HFEN}_k(\mathbf{a}, \mathbf{b}) = \frac{||\mathcal{G}(\mathbf{a}) - \mathcal{G}(\mathbf{b})||_k}{||\mathcal{G}(\mathbf{b})||_k}, \tag{12}$$

where $\mathcal{G}$ is a Laplacian-of-Gaussian filter (Ravishankar and Bresler, 2011) with kernel of size $15 \times 15$ and with a standard deviation of 2.5, and $k = 1$ or 2.

– Mean Average Error (MAE / $L_1$) Loss

$$\mathcal{L}_1(\mathbf{a}, \mathbf{b}) = ||\mathbf{a} - \mathbf{b}||_1 = \sum_{i=1}^{n} |a_i - b_i| \tag{13}$$

- $k$-space Domain Loss Functions
  - Normalized Mean Squared Error (NMSE)

$$\mathcal{L}_{\text{NMSE}} := \text{NMSE}, \quad \text{NMSE}(\mathbf{a}, \mathbf{b}) = \frac{||\mathbf{a} - \mathbf{b}||_2^2}{||\mathbf{a}||_2^2} = \frac{\sum_{i=1}^{n}(a_i - b_i)^2}{\sum_{i=1}^{n} a_i^2}. \tag{14}$$

  - Normalized Mean Average Error (NMAE)

$$\mathcal{L}_{\text{NMAE}} := \text{NMAE}, \quad \text{NMAE}(\mathbf{a}, \mathbf{b}) = \frac{||\mathbf{a} - \mathbf{b}||_1}{||\mathbf{a}||_1} = \frac{\sum_{i=1}^{n} |a_i - b_i|}{\sum_{i=1}^{n} |a_i|}. \tag{15}$$

The rationale for the loss function components is also drawn from the literature (Yiasemis et al., 2025). In the frequency domain, $\mathcal{L}_{\text{NMSE}}$ and $\mathcal{L}_{\text{NMAE}}$ are used to evaluate global similarity to the fully sampled $k$-space, with the former addressing larger deviations and the latter focusing on finer discrepancies. In the image domain, $\mathcal{L}_1$ and $\mathcal{L}_{\text{SSIM}}$ are commonly combined to optimize pixel-level accuracy and perceptual quality, while $\mathcal{L}_{\text{HFEN}_k}$ emphasizes the preservation of edges and fine details.

## Appendix D. Supplementary Experiments

### D.1. Robustness to Model Choice Experiments

Here, we present supplementary experiments aimed at further validating the efficacy of our proposed JSSL method. These experiments involve a comparative analysis between JSSL and traditional SSL MRI Reconstruction. We adapt the methodologies outlined in Section 3 of our main paper, utilizing two distinct deep MRI reconstruction models instead of the vSHARP architecture:

- Utilizing a plain image domain U-Net (Ronneberger et al., 2015), a non-physics-based model that takes an undersampled-reconstructed image as input and refines it. Specifically, we employ a U-Net with four scales and 64 filters in the first channel.
- Employing an End-to-end Variational Network (E2EVarNet) (Sriram et al., 2020), a physics-based model that executes a gradient descent-like optimization scheme in the $k$-space domain. For E2EVarNet, we perform 6 optimization steps using U-Nets with four scales and 16 filters in the first scale.

To estimate sensitivity maps for both architectures, an identical Sensitivity Map Estimation (SME) module was integrated, mirroring the experimental setup outlined in our primary paper.

Both models underwent training and evaluation on data subsampled with acceleration factors of 4, 8, and 16, with ACS ratios of 8%, 4%, and 2% of the data shape. Choices of hyperparameters for JSSL and SSL are the same as in the comparative experiments presented in Section 3.6. Additionally, choices for proxy and target datasets, as well as data splits, are also the same as in the main paper.

Experimental setups were executed on NVIDIA A100 80GB GPUs, utilizing 2 GPUs for U-Net and 1 GPU for E2EVarNet. We employed batch sizes of 2 and 4 for U-Net and E2EVarNet, respectively, on each GPU. The optimization procedures, initial learning rates, and the employed optimizers aligned with those utilized in the main paper.

Table S2: Model architectures parameters.

| Model | Parameter Count (millions) | Physics Model | Training Iterations (k) | Learning Rate Reduction Schedule | Inference Time (s) per volume |
|---|---|---|---|---|---|
| vSHARP | 95 | ADMM | 700 | 150k | 17.7 |
| U-Net | 33 | - | 375 | 75k | 13.1 |
| E2EVarNet | 13.5 | Gradient Descent in $k$-space Domain | 250 | 50k | 13.9 |

Table S2 details the model specifics for all considered architectures presented in both the main paper and this section.

#### D.1.1. ROBUSTNESS TO MODEL CHOICE EXPERIMENTS RESULTS

The average results of our supplementary comparative studies to assess JSSL's robustness to different architecture choices are provided in Table S3. From these results we observe alignment with our original findings: JSSL-trained models consistently outperform SSL-trained models for both architecture choices.

Table S3: Robustness to model choice experiments results. An asterisk ($*$) denotes that the average best method (bold) was not found to be statistically significantly better than the corresponding method ($p > 0.05$).

| Architecture | Setup | 4x | | | 8x | | | 16x | | |
|---|---|---|---|---|---|---|---|---|---|---|
| | | SSIM | pSNR | NMSE | SSIM | pSNR | NMSE | SSIM | pSNR | NMSE |
| U-Net | SSL | $0.854_{\pm 0.031}$ | $33.0_{\pm 1.6}$ | $0.013_{\pm 0.004}$ | $0.742_{\pm 0.040}$ | $29.4_{\pm 1.4}$ | $0.030_{\pm 0.006}$ | $0.651_{\pm 0.051}$ | $\mathbf{26.7}_{\pm 1.5}$ | $\mathbf{0.055}_{\pm 0.009}$ |
| | JSSL | $\mathbf{0.863}_{\pm 0.031}$ | $\mathbf{33.5}_{\pm 1.5}$ | $\mathbf{0.012}_{\pm 0.002}$ | $\mathbf{0.759}_{\pm 0.042}$ | $\mathbf{29.7}_{\pm 1.4}$ | $\mathbf{0.027}_{\pm 0.005}$ | $\mathbf{0.663}_{\pm 0.051}$ | $26.7_{\pm 1.4}^{*}$ | $0.055_{\pm 0.009}^{*}$ |
| E2EVarNet | SSL | $0.874_{\pm 0.029}$ | $33.7_{\pm 1.7}$ | $0.011_{\pm 0.003}$ | $0.770_{\pm 0.039}$ | $30.0_{\pm 1.4}$ | $0.025_{\pm 0.006}$ | $0.670_{\pm 0.051}$ | $27.0_{\pm 1.5}$ | $0.051_{\pm 0.009}$ |
| | JSSL | $\mathbf{0.888}_{\pm 0.032}$ | $\mathbf{34.9}_{\pm 1.6}$ | $\mathbf{0.008}_{\pm 0.002}$ | $\mathbf{0.784}_{\pm 0.042}$ | $\mathbf{30.5}_{\pm 1.4}$ | $\mathbf{0.023}_{\pm 0.005}$ | $\mathbf{0.678}_{\pm 0.053}$ | $\mathbf{27.1}_{\pm 1.5}$ | $\mathbf{0.050}_{\pm 0.009}$ |

Furthermore, the superior performance of vSHARP and E2EVarNet compared to the U-Net model in both SSL and JSSL settings across all acceleration factors highlights the advantage of adopting physics-guided unrolled models for reconstruction. It is also worth mentioning that vSHARP consistently outperformed E2EVarNet at all accelerations.

# Appendix E. Additional Figures

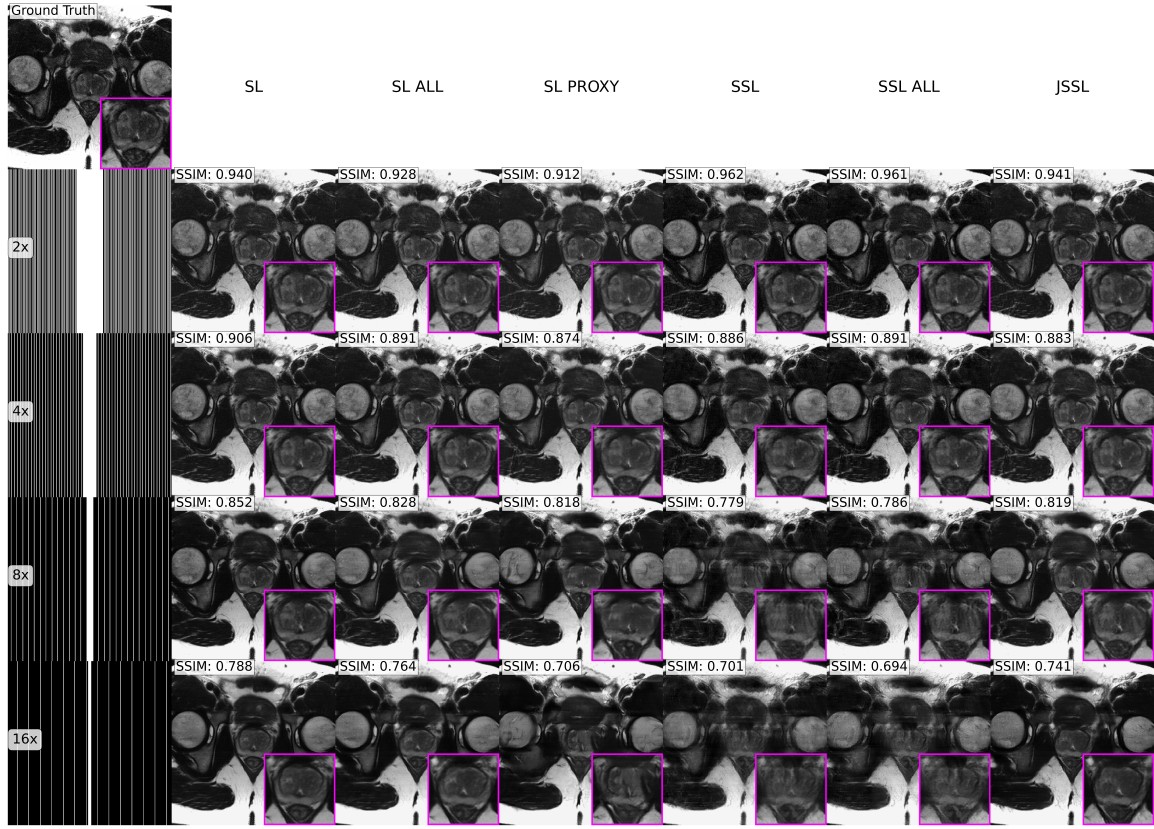

Figure S2: Example reconstructions of a prostate MRI slice subsampled at different acceleration factors from the test set in experiment set **A**. Each training setup is compared against the ground truth. The zoomed-in region highlights the prostate.

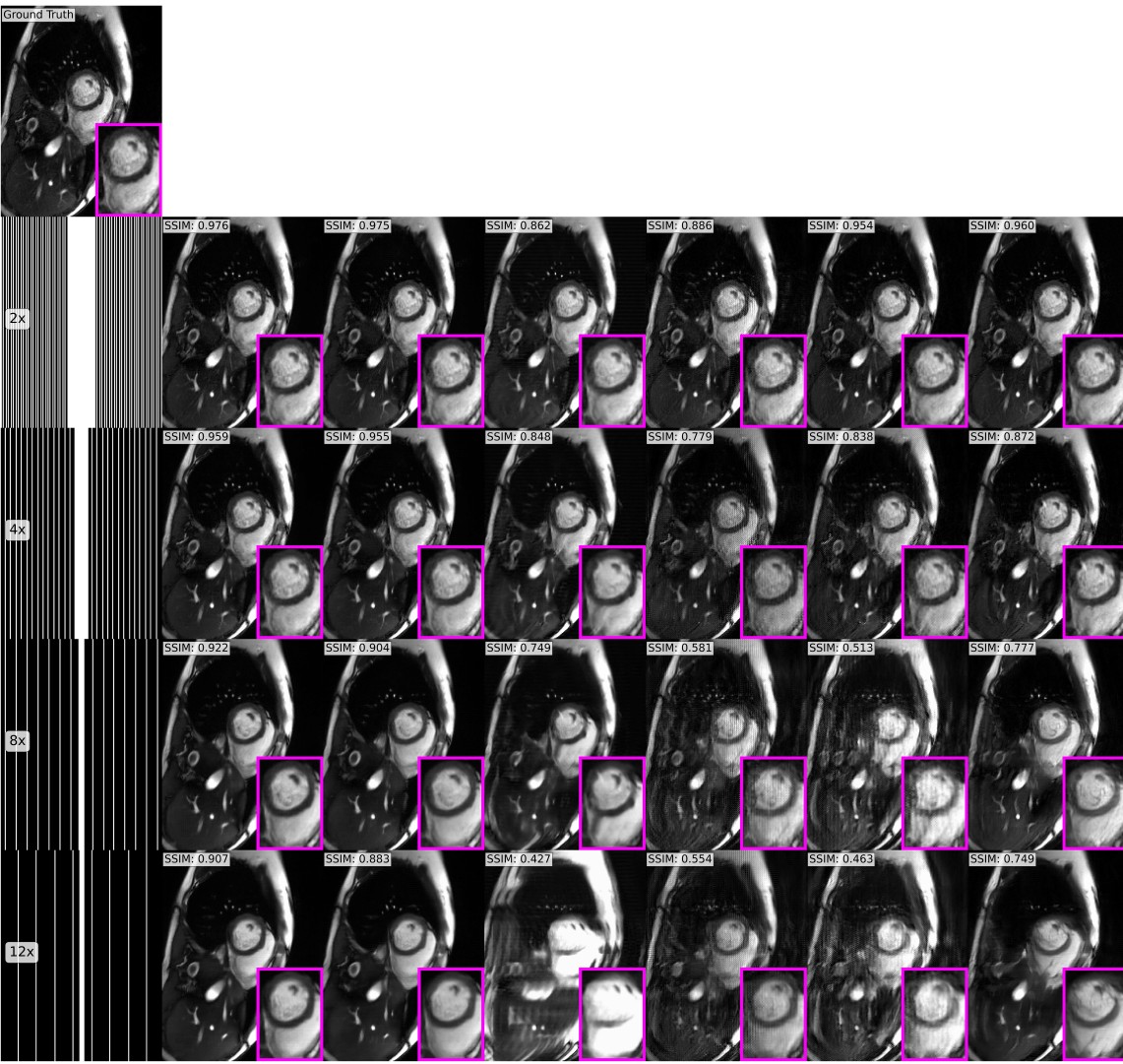

Figure S3: Example reconstructions of a cardiac MRI slice subsampled at different acceleration factors from the test set in experiment set **B**. Each training setup is visualized against the ground truth. The zoomed-in region highlights the heart.

## Appendix F.  Extended Discussion

### F.1.  Limitations

While our experiments indicate that JSSL demonstrates improvements over conventional SSL methods, several limitations warrant discussion. Firstly, the efficacy of JSSL is highly dependent on the availability and quality of proxy datasets. Although datasets such as the fastMRI datasets contain fully-sampled data and are readily available, there might be instances where such datasets cannot be used. This could occur in cases where the anatomical regions of interest in the proxy datasets are not sufficiently similar to those in the target dataset, or where differences in imaging protocols and acquisition parameters introduce significant discrepancies.

For instance, in experiment set **A**, where the fastMRI prostate data served as the target domain and brain and knee fastMRI datasets were used as proxies, the SL PROXY setup showed relatively good performance, indicating that training with similar proxy domains can still be beneficial for out-of-distribution inference. However, in experiment set **B**, where the CMRxRecon cardiac data was the target and brain, knee, and prostate fastMRI datasets served as proxies, the performance of SL PROXY was significantly lower than all methods, highlighting that when proxies are dissimilar to the target, SL PROXY struggles to generalize effectively. In both scenarios, JSSL consistently surpassed SL PROXY, indicating that the combined supervised and self-supervised approach is more robust, regardless of the proxy dataset's similarity.

Additionally, the inclusion of proxy datasets in training can introduce biases, particularly if there are substantial differences between the proxy and target domains. This bias could potentially degrade the model's performance on the target dataset, as observed in some of our supervised learning experiments.

Moreover, similar to any DL-based method, JSSL's performance is influenced by the choice of loss functions for each component of the JSSL loss and their weighting in the loss $\mathcal{L}_{\psi}^{\text{JSSL}}$. In our experiments, we employed identical dual-domain loss functions for each component and equal weighting for the SL and SSL components (see Appendix Section C). However, different loss and weighting choices might affect JSSL's performance.

JSSL performance also depends on the partitioning strategy used for subsampled data in self-supervised learning. While we adopted a Gaussian partitioning scheme, alternative strategies might yield different results and require further exploration. The optimal partitioning scheme may vary depending on the specific characteristics of the target and proxy datasets, as well as the desired reconstruction quality.

Lastly, our experiments are limited to comparing only one SSL method (SSDU) and does not consider other proposed self-supervised methodologies. However, the reason for comparing to SSDU only is that we consider it representative, as most SSL-based methods are derivatives of SSDU and still employ SSL-based losses to train their models (refer to Apendix A). In addition, comparing to methods that train more than one model as their SSL task is outside the scope of this research, as this can introduce additional computational difficulties and are derivatives of the SSDU method. Our purpose is to compare JSSL and SSL training methods in their general forms.

