# OpenReview forum: "Joint Supervised and Self-supervised Learning for MRI Reconstruction"
_MIDL.io/2025/Conference — MIDL 2025 Poster_

### Official Review · Reviewer_CKSs · 2025-02-18

**Confidence:** 3
**Preliminary Rating:** 5
**Recommendation:** Best Paper Award, Oral
**Final Rating:** 5

**Summary:**

This paper introduces a method leveraging deep learning for reconstructing MRI data from undersampled inputs. The approach combines self-supervised learning in target domains (lacking fully sampled data) with supervised learning in proxy domains (where fully sampled data is available). The authors provide comprehensive mathematical details and compelling results across two experimental settings. Additionally, they offer practical guidance on choosing training methods for MRI reconstruction.

**Strengths:**

- Addressing a crucial area in MRI research and clinical applications.
- Detailed method description bolstered by solid motivation.
- Convincing experimental results.
- Architectural flexibility of the proposed method.
- Useful guidelines for practitioners.

**Weaknesses:**

- No significant weaknesses identified.
- The main text relies on the Appendix for full understanding, limiting accessibility.
- More intuitive explanations, especially in Section 2.3 (MRI Reconstruction), would enhance readability.
- Shortcomings are primarily discussed in the appendix; a brief summary in the main text would improve clarity.

**Detailed Comments:**

- **Section 2.5 - Coil Sensitivity Prediction**: Clarification needed on where this is illustrated in Figure 1.
- **Section 3.1 - Datasets**: What motivated the choice of three proxy domains in Experiment B instead of two?
- **Section 3.2 - Subsampling Schemes**: Did the authors consider or experiment with other subsampling strategies? How easily could the proposed method adapt to these scenarios?
- **Section 4 - Discussion and Conclusion**: It would be beneficial to include a concise summary of the main limitations in the main text.

**Justification Of The Final Rating:**

I recommend to accept this paper due to its substantial contributions to MRI reconstruction through an integration of supervised and self-supervised learning techniques. The authors effectively address a pivotal challenge in medical imaging: reconstructing high-quality images from undersampled MRI data, which has crucial implications for both research and clinical practice. Their mathematical framework, coupled with thorough experimental validation, demonstrates the benefits of the proposed approach. It can also serve as a good branching point for further developments. Although some areas could benefit from greater clarity, the main contributions are well-articulated, and the importance of this work makes it a strong candidate for acceptance. The authors have successfully addressed feedback, enhancing the manuscript's self-containment, making it particularly suitable for publication in a potential special issue.

**Justification Of The Preliminary Rating:**

I strongly recommend accepting this paper due to its significant contributions to the field of MRI reconstruction through the innovative integration of supervised and self-supervised learning approaches. The methodology proposed addresses a critical challenge in medical imaging, specifically the issue of reconstructing high-quality images from undersampled MRI data, which has substantial implications for both research and clinical practice. The authors provide a thorough mathematical framework and robust experimental validation, demonstrating the effectiveness of their approach across various settings and datasets.

The results presented are compelling, showcasing improved reconstruction quality compared to existing methods. Additionally, the flexibility of the proposed architecture for different MRI tasks suggests its wide applicability. The practical insights offered in the guidelines for training strategies further enhance the paper's value for practitioners in the field. While some sections could benefit from clearer explanations, the overall clarity of the main contributions, combined with the importance of the topic and the authors' thorough approach, make this work a strong candidate for acceptance. Overall, the paper presents novel insights that can significantly advance the field of MRI reconstruction.

**Questions To Address In The Rebuttal:**

See the comments above

**Special Issue:**

Yes

---

> ### Author Response · Authors · 2025-03-06
> **Responses to Reviewer CKSs's comments**
>
> We sincerely thank you for your highly positive and encouraging feedback. We deeply appreciate your thoughtful review, your recognition of our contributions, and your recommendation for a Best Paper Award and MELBA Special Issue. Your insightful comments have been invaluable in refining our work, and we are truly grateful for your time and effort in evaluating our manuscript. Below, we address your suggestions and outline the corresponding revisions.
>
>
> 1. **[Appendix Dependence]**
> We acknowledge the initial reliance on the appendix for key details and have revised the manuscript to enhance self-containment and readability:
>     - A summary of related work is now included in the Introduction, while an extended discussion remains in Appendix A.
>     - All equations in Section 2 are now explicitly defined within the main text.
>     - Implementation details (Section 3.3) and evaluation methods (Section 3.5) have been integrated into the main text.
>     - A discussion of method limitations has been added to Section 4, with further details in the appendix.
>
> 2. **[Coil Sensitivity Prediction]**
> To avoid overcrowding Figure 1, sensitivity maps were not explicitly illustrated. However, they are inherently included in the reduce  and expand operators  ($\mathcal{R}_S, \mathcal{E}_S $), as they are computed at the start of the reconstruction pipeline. Specifically, they are extracted from autocalibration signal data in both SL and SSL steps, processed via the sensitivity map predictor (SMP), and used within these operators. We have added a clarifying caption to Figure 1.
>
> 3. **[Datasets]**
> We carefully designed two experimental setups to evaluate JSSL across different conditions:
>     - Experiment Set **A**: The prostate dataset (target) was paired with knee and brain datasets (proxies) from the same vendor (fastMRI). This setting allowed us to analyze JSSL’s performance when proxy datasets share vendor-specific characteristics, despite anatomical differences.
>     - Experiment Set **B**: The cardiac dataset (target) from CMRxRecon 2023 was paired with the fastMRI datasets (knee, brain, prostate) as proxies. This scenario tested JSSL’s adaptability when proxy datasets differed significantly in both anatomy and acquisition source (vendor/scanner).
>
> 4. **[Subsampling schemes]**
> The retrospective subsampling strategies were chosen based on dataset-specific considerations:
>     - Brain and knee MRI: We used the 1D Cartesian schemes provided in the fastMRI challenge—random uniform for brain and equispaced for knee.
>     - Cardiac MRI: We followed the equispaced scheme used in the CMRxRecon 2023 challenge.
>     - Prostate MRI: The equispaced scheme was selected to match the prospective sampling strategy used in data acquisition.
> While we adopted these schemes based on dataset conventions, our method is not inherently tied to any particular subsampling strategy. JSSL should generalize across different sampling schemes and their combinations in proxy/target datasets. However, we acknowledge that the choice of partitioning strategy in SSL/JSSL could impact performance. In our experiments, we employed a Gaussian partitioning strategy (see Appendix C.2), which has demonstrated superior performance in SSL methods. Further exploration of its impact on JSSL performance could be conducted in future work.
>
> 5. **[Discussion and Conclusion]**
> In response to your suggestion, we have added a concise summary of the method’s limitations in the Discussion and Conclusion section while keeping an extended discussion in the appendix.
>
>
> Once again, we are truly grateful for your incredibly positive assessment of our work. Your recognition of our contributions, as well as your recommendation for a Best Paper Award and MELBA Special Issue, is an immense honor. It is rewarding to know that our approach is considered valuable for MRI reconstruction and clinical applications. We deeply appreciate your constructive feedback, which has helped us further refine the manuscript.

---

> > ### Comment · Reviewer_CKSs · 2025-03-12
> >
> > Thanks so much for revising your work and addressing my comments. I think this is a strong paper.

---

> > ### Author Response · Authors · 2025-03-12
> > **Appreciation for Their Review and Final Comments to Reviewer CKSs**
> >
> > Thank you for your kind words and for acknowledging our revisions. We truly appreciate your thoughtful feedback and the time you took to review our work. Your strong accept recommendation and support for a Special Issue mean a lot to us, as we believe this work will be valuable to the community.

---

### Official Review · Reviewer_huZP · 2025-02-18

**Confidence:** 5
**Preliminary Rating:** 1
**Recommendation:** Poster
**Final Rating:** 2

**Summary:**

The paper proposes what the authors call joint supervised and self-supervised learning for MRI reconstruction. The main idea is to train the reconstruction model in an SSL manner using undersampled data from the target dataset and in a supervised manner using fully sampled data from proxy datasets. The authors validate their work using the fastMRI and CMRxRecon 2023 datasets. The conclusion is that the proposal can improve results with a fully supervised learning approach being an upper-bound on model reconstruction performance.

**Strengths:**

The paper's idea is simple, and it makes a lot of sense to train reconstruction models using both fully sampled and undersampled data since, in practice, the goal is to no longer collect fully sampled data in the future and have all scans accelerated.

**Weaknesses:**

The paper is hard to read. A lot of the details and discussion have been pushed to the appendices.  It has too many mathematical equations to describe a straightforward idea, which has limited novelty. Semi-supervised approaches might be new in MRI recon, but they have been explored in many other fields, including medical imaging.

**Detailed Comments:**

- I would suggest simplifying the technical explanations in the paper. For equations, make sure all terms are defined in the text.
- I would suggest showing only key results in the main body of the manuscript. The reconstruction panels are too crowded, and the quality of the MRI reconstructions is hard to appreciate.
- What would happen if one dataset corresponds to a 2D acquisition and the other is 3D? Can you mix 1D and 2D sampling schemes with the proposed methodology?

**Justification Of The Final Rating:**

I appreciate the authors' effort in improving the manuscript. I believe the manuscript has improved, but only slightly, which is the reason I only increased my score to "weak reject".

Section 2 ("Materials and Methods"), in my view, is still overly complicated to explain simple ideas and concepts. I also don't think it covers the "materials" since the datasets are described in section 3 (?).

There is still too much information in the main paper. Many tables and figures are not legible at 100% magnification. Why not stick to R = {4,8} results in the main paper?

- Figure 1 is not fully legible at 100% zoom.
- In Table 2, the std values are not legible at 100% zoom.
- Figure 2 - There are 25 reconstructions shown in a mosaic, and it is extremely difficult to appreciate the reconstruction quality.

**Justification Of The Preliminary Rating:**

The proposal has limited novelty, and the paper's presentation is unclear, relying too much on its appendix, making it hard to appreciate any potential findings that can be drawn from the experiments presented by the authors.

**Questions To Address In The Rebuttal:**

I think the paper relies too heavily on its supplementary material. In my opinion, a complete restructuring would be needed to provide a compelling and clear story of the proposed method and results.

I would suggest simplifying the presentation of the paper idea, which is simple, and focusing more on the different experiments and what can be learned from them.

---

> ### Author Response · Authors · 2025-03-06
> **Responses to Reviewer huZP's comments**
>
> We sincerely appreciate the time and effort you have taken to review our work. While we acknowledge your concerns, we also strongly believe that our approach presents a meaningful contribution to the field of MRI reconstruction. Your detailed feedback has been extremely helpful in refining our manuscript, and we have worked diligently to improve clarity, readability, and presentation based on your suggestions.
>
> 1. **[Definition of Terms]**
> All terms are now clearly defined in the main text to enhance readability and ensure clarity.
>
> 2. **[Reconstruction Images]**
> We have selected more representative reconstruction examples demonstrating the advantages of JSSL in realistic clinical scenarios, and improved contrast to enhance visibility. Additionally, we have reduced the size of SSIM labels and carefully chosen zoomed-in regions to improve the clarity of image comparisons.
>
> 3. **[Mixing 2D and 3D Dataset Acquisitions]**
> The proposed JSSL methodology, like SSL or SL approaches, is not inherently constrained by acquisition dimensionality. This primarily depends on the configuration of the reconstruction model rather than the training setup. While all datasets used in our experiments consist of 2D acquisitions, our approach is equally applicable to 3D acquisitions.
>
> 4. **[Mixing 1D and 2D Sampling Schemes]**
> JSSL is agnostic to the sampling scheme, meaning that mixing 1D and 2D schemes should be feasible. Previous work [1] in supervised learning has demonstrated that combining different sampling schemes can enhance generalizability. We would expect similar benefits in JSSL.
>
> 5. **[Novelty]**
> We respectfully disagree with the comment regarding limited novelty. To our knowledge, this is the first work to propose a method that simultaneously trains a model in both an SSL setting (using undersampled data from target datasets) and an SL setting (leveraging fully sampled proxy datasets) when fully sampled target dataset measurements are unavailable. Our methodology is thoroughly explained in Section 1 of the main paper and Appendix A, highlighting its distinctions from prior work. Furthermore, we provide theoretical justification for our approach and conduct a comprehensive evaluation across two experimental settings.
> JSSL is also a natural and practical solution given the current state of publicly available MRI datasets. Fully sampled data already exist for certain organ sites through public reconstruction challenges, while for others, such datasets are unavailable. The real question is how to best leverage the available combination of fully sampled and undersampled data, which our method directly addresses.
> Additionally, JSSL offers a computational advantage over SSL-only approaches. Instead of investing substantial computational resources into training or designing highly complex SSL-specific architectures, our method efficiently utilizes fully sampled proxy data to accelerate training and improve reconstruction quality, making it a more scalable and practical alternative.
> While the concept may appear intuitive, its novelty and potential impact on accelerated MRI reconstruction remain significant. Given the opportunity to gain broader recognition within the community, JSSL has the potential to reshape the way MRI reconstruction models are trained, balancing efficiency, performance, and real-world applicability.
>
> 6. **[Reliance on Supplementary Material]**
> We acknowledge the initial reliance on supplementary material and have restructured the manuscript to provide a more self-contained and cohesive narrative:
>     - A summary of related work is now included in the Introduction, with an extended version in Appendix A.
>     - All equations in Section 2 are now explicitly defined within the main text.
>     - We have minimized excessive references between sections and appendices by consolidating redundant information. Some key details previously in the appendices have been moved to the main text for better accessibility.
>     - Details on implementation and optimization (Section 3.3.) and evaluation (Section 3.5.) are now integrated into the main paper.
>     - A discussion on method limitations is now included in Section 4, with further details in the appendix.
>
> We deeply value your feedback and have made substantial efforts to improve the paper in response to your concerns. Given these revisions, we sincerely hope you now recognize the significance and practical impact of our work.
> This paper represents an important contribution to the MRI reconstruction community, and we strongly believe that its acceptance will benefit both researchers and clinicians working toward accelerated MRI solutions. We hope that with these improvements, you will now see its merit and support its publication. Once again, thank you for helping us make our paper even stronger!
>
> References
> ---------------
> [1] Yiasemis, G, et al. "On retrospective k-space subsampling schemes for deep MRI reconstruction." Magnetic Resonance Imaging 107 (2024): 33-46.

---

> ### Author Response · Authors · 2025-03-13
> **Response to Final Rating and Justification of Reviewer huZP**
>
> We sincerely appreciate the reviewer’s continued feedback and their acknowledgment of the improvements made to our manuscript. Below, we address the remaining concerns raised in the updated review, as reflected in the Final Rating and Justification.
>   1. **[Clarity of Section 2 ("Materials and Methods")]**
> We understand the reviewer’s concern regarding the complexity of Section 2. However, the mathematical formulations, particularly in the loss definitions, are fundamental to providing a precise and reproducible description of our methodology. They are essential for ensuring clarity in implementation and for properly defining the JSSL training process. Additionally, further reducing content in this section or moving it to the appendix would contradict the original concerns regarding over-reliance on supplementary material.
>
>   2. **[Placement of Dataset Descriptions]**
> We acknowledge that the dataset descriptions are currently located in Section 3 rather than Section 2. If restructuring this section enhances clarity, we are open to making this adjustment upon acceptance.
>
>   3. **[Legibility of Figures and Tables]**
> While our tables and figures are clearly legible in standard A4 print form, we recognize that digital readability at 100% zoom is also important. On our devices, the images appear sufficiently clear at 100% zoom, and since the digital format allows for zooming, we provide high-DPI images to maintain clarity. Nevertheless, to further enhance readability, we will:
>      -  Increase the font size of standard deviation values in tables.
>      -  Ensure that all figures are rendered at the highest possible quality.
>      -  Improve Figure 1 by potentially enlarging the zoomed-in region, particularly to enhance the interpretability of lesion-focused visualizations.
>
>   4. **[Presentation of Reconstruction Results (Figure 2)]**
> The reviewer notes difficulty in assessing reconstruction quality due to the “mosaic” layout in Figure 2. However, the inclusion of zoomed-in regions was intentional, as it allows for a clearer evaluation of subtle reconstruction details, particularly for cardiac structures. We believe that presenting reconstructions for all methods is crucial to ensuring a fair comparison.
> Additionally, we included results at higher acceleration factors ($R = 12, 16$) because they are relevant for ultra-fast imaging protocols, where extreme undersampling is often necessary to meet clinical time constraints. For example, in real-time cardiac MRI, higher acceleration factors are essential for capturing transient physiological events without compromising temporal resolution. That said, we acknowledge the reviewer’s suggestion to focus on middle acceleration factors ($R = 4, 8$) in the main paper, as this would improve readability while maintaining a balanced presentation of results. Upon acceptance, we are willing to implement this change while keeping the full set of results available in the appendix for completeness.
>
> Given these clarifications, we sincerely hope the reviewer recognizes the substantial improvements we have made. If any of the above responses help address their remaining concerns, we would greatly appreciate further reconsideration of the rating. Regardless, we remain grateful for the constructive feedback and the opportunity to further refine our work.

---

> > ### Comment · Area_Chair_vhuy · 2025-03-14
> > **multiple steps of rebuttal**
> >
> > We appreciate if reviewers can double check if authors sufficiently addressed the comments at each round of the rebuttal and update their score if they decide that there is sufficient change in their overall evaluation of the paper.
> >
> > Area Chair

---

### Official Review · Reviewer_emRD · 2025-02-20

**Confidence:** 4
**Preliminary Rating:** 2
**Recommendation:** Poster
**Final Rating:** 4

**Summary:**

MR Imaging is slow, but its acceleration via reducing the k-space acquisition leads to images with alias and artefacts. The problem is even more prominent when moving organs are imaged. This paper presents a joint supervised (SL) and self supervised (SSL) learning approach to overcome some limitation of previous reconstruction methods.

**Strengths:**

- The topic is really interesting and relevant to MIDL, the contribution is valuable.
- The authors combine a self supervised with a supervised approach. The method leverages the availability of proxy fully sampled datasets in conjunction with subsampled data.

**Weaknesses:**

- Reading this paper is painful, requiring continuous hopping between sections within the main document and/or the appendix.
- The paper would benefit from a more comprehensive focus on the reconstruction of features which are clinically important.

**Detailed Comments:**

- Due to "Appendix A reviews related work on SSL-based MRI reconstruction, providing the background for our approach.", Appendix A could be brought into the main manuscript. I understand there is a page limit, perhaps most of the Introduction section could be removed, since it's content, well known to readers of the paper.
- In equation 4, DC is not defined, it's defined in the appendix. Similar to the Fourier operator F $\epsilon_{s}$.

**Justification Of The Final Rating:**

I believe the authors have done a great job with improving the paper readability and I also appreciate the fact they took feedback from all the reviewers very seriously. I find this paper interesting and I would like to upgrade my rating.

**Justification Of The Preliminary Rating:**

The topic of the paper is interesting, but its exposition could be improved otherwise I doubt readers will benefit from findings of this work. The content is focused mostly on the technical explanation of the approach and then it falls short when it comes discussing results especially on the analysis of the (visual) quality of the reconstructed images from a clinical standpoint.

**Questions To Address In The Rebuttal:**

Could the authors comment on the artifacts they observed in the reconstructions? What implications and risks they foresee if these artifacts were not caught by the quantitative metrics? Were these obvious artifacts that clinical readers could easily spot, or were these subtle ones which could potentially lead to mis-dignosis?

**Special Issue:**

No

---

> ### Author Response · Authors · 2025-03-06
> **Responses to Reviewer emRD's comments**
>
> We sincerely appreciate your thoughtful and constructive feedback. Your recognition of the significance and relevance of our contribution is highly encouraging, and we have carefully addressed your concerns to enhance the clarity and impact of our manuscript. Below, we outline the key revisions we have made based on your suggestions.
>
> 1. **[Related Work and Background]**
> We have incorporated more information on related work and background for JSSL in the introduction, while retaining an extended discussion in Appendix A.
>
> 2. **[Operators Definitions]**
> To improve clarity, we now explicitly define all operators, including the DC and encoding operator $\mathcal{E}_{S}$ , within the main paper.
>
> 3. **[Paper Presentation and Readability]**
> We have made several modifications to enhance readability and improve the overall presentation of the paper:
>     - Mathematical Definitions: All mathematical definitions of forward, backward, and other relevant operators are now included in the main paper (Section 2.1).
>     - Descriptive Comments: Additional explanations are provided where relevant to improve comprehension.
>     - Reduction of Cross-Referencing: We have minimized excessive references between sections and appendices by consolidating redundant information. Some key details previously in the appendices have been moved to the main text for better accessibility.
>     - Simplified Equations: We have removed redundant mathematical equations from Sections 2.2 and 2.3 while preserving all necessary context.
>     - Appendix Content Relocation: Relevant information has been moved from the appendices to the main paper, particularly into Section 3.
>     - Clinically Relevant Examples: We have updated Figures 2 and 3 to include more clinically meaningful reconstruction examples, providing a better assessment of the method’s impact on medical imaging.
>
> 4.	**[Reconstruction Artifacts Analysis]**
> Our quantitative results (Tables 1 and 2) show that the proposed JSSL method outperforms SSL methods in reconstruction quality, while supervised methods achieve the best performance. However, our focus is not solely on maximizing quantitative performance but rather on demonstrating the advantages of JSSL in realistic clinical scenarios.
> To supplement the quantitative evaluations, we now provide more relevant qualitative results in Figures 2 and 3 (main paper) and additional reconstructions in Figures S2 and S3 (Appendix E). These include center crops of key regions of interest, specifically:
>     - Prostate (Experiment Set A): We focus on the area surrounding the prostate. Notably, Figure 2 includes reconstructions from a case where the ground truth MRI was labeled PIRADS 4 (indicating likely clinically significant cancer). This allows for a meaningful assessment of lesion visibility across the different training setups.
>     - Cardiac Data (Experiment Set B): We emphasize the heart region, which is critical for clinical evaluation. The reconstructions allow visualization of the heart’s shape and cardiac phase across different acceleration factors.
>
>     **Artifact Analysis:**
>     - JSSL reconstructions exhibit a clear trend consistent with the quantitative metrics, producing images closer to the ground truth than SSL.
>     - As expected, image quality degrades as acceleration increases (for all methods, including supervised ones) and images become more noisy.
>     - At high accelerations, SSL reconstructions exhibit blurring and oversmoothing, particularly in the prostate region, where fine details are lost. In contrast, JSSL preserves structural details better, though some smoothing is still observed at extreme acceleration factors (e.g., 16×).
>     - In the prostate cancer case (Figure 2), the lesion (indicated by the arrow and bounding boxes) remains distinguishable across JSSL reconstructions at all acceleration factors.
>     - In prostate reconstructions (Figures 2, S2), JSSL maintains the prostate shape even at high acceleration, which is not the case for SSL.
>     - In cardiac reconstructions (Figures 3, S3), JSSL maintains the heart’s shape and phase visibility at all acceleration levels. However, at high accelerations, minor ghosting and blurring artifacts appear. Additionally, less clinically significant structures, such as ribs, become indistinguishable.
>     - SSL cardiac reconstructions show more pronounced artifacts, such as ringing, at high acceleration factors.
> A discussion on these findings has been added in the revised manuscript, including updated figure captions and a more detailed Discussion and Conclusion section (Section 4).
> We sincerely appreciate your time and constructive feedback, which has helped us significantly improve the clarity and clinical focus of our paper. We hope these revisions now make the contributions of JSSL more accessible and compelling, and that you recognize the value it brings to the field of MRI reconstruction.
>
> Thank you again for your thoughtful review and for helping us refine this work.

---

> ### Author Response · Authors · 2025-03-12
> **Appreciation for Their Thoughtful Review and Updated Rating to Reviewer emRD**
>
> We sincerely appreciate the time and effort you invested in reviewing our paper and providing constructive feedback.
>
> We are especially grateful for your recognition of our revisions and your decision to update your rating. Your careful reading and insightful comments played a crucial role in enhancing the final quality of the paper.
>
> Thank you again for your valuable input and for contributing to the improvement of our work!

---

### Author Rebuttal · Authors · 2025-03-06

**Rebuttal:**

We sincerely appreciate the valuable feedback from all three reviewers. Their insights have significantly improved the manuscript, and we have carefully addressed all comments in our revisions. We now believe the paper is ready for acceptance following these changes and your thoughtful input.

For the rebuttal, we have uploaded a single zip file containing:

- "Revised Manuscript.pdf" – The updated version of the paper.
- "Revised Manuscript with Highlighted Changes.pdf" – The revised manuscript with all modifications clearly marked.
- "Responses to Reviewers.pdf" – A detailed document addressing each reviewer's comments individually.

Additionally, we have responded to each reviewer’s comments directly using the Official Comment feature in the review system.

Thank you again for your time and constructive feedback.

**Supporting Material:**

/attachment/032b3428421a6ad6976a4c127f41e05d9812c0cd.zip

---

### Author Response · Authors · 2025-03-14
**Concern Regarding Incomplete Discussion in Review Process**

Dear Program Chairs/Area Chairs,

We would like to draw your attention to the incomplete discussion regarding Reviewer **huZP**'s comments on our submission. While we greatly appreciate the reviewer's feedback and the revision process, we are concerned that the discussion phase did not fully serve its intended purpose.

Despite our substantial efforts in addressing the reviewer's concerns (as also acknowledged by Reviewers **emRD** and **CKSs**), Reviewer **huZP** did not engage in further discussion. Instead, they directly issued a final rating adjustment (from strong reject to weak reject), accompanied by comments that we could have readily addressed had there been an opportunity for further dialogue.

Given that the discussion phase is designed for clarifications and additional responses, we believe that the remaining concerns raised by Reviewer **huZP** could have been effectively resolved in the final version, as outlined in our responses. However, we did not receive any further engagement from the reviewer after our rebuttals.

We understand that reviewer engagement is voluntary, but given the structure of the discussion period, we wanted to bring this to your attention, as it may impact the fairness of the review process.

Thank you for your time and consideration!

---

> ### Comment · Area_Chair_vhuy · 2025-03-14
> **discussions during rebuttal phase**
>
> Dear Authors,
>
> We appreciate your efforts to address the concerns of the reviewers in the rebuttal period. I see that both of your reviewers updated their scores after your efforts during the rebuttal. I have observed that Reviewer huZP has responded after your initial rebuttal and updated his score. I will send huZP a reminder to check if he/she wants to update the score after your second response.
>
> Best regards,
> Area Chair.

---

> > ### Author Response · Authors · 2025-03-14
> > **Discussions During Rebuttal Phase**
> >
> > Thank you for your response and for following up with Reviewer huZP.
> >
> > We really appreciate your time and effort in overseeing the process.

---

### Meta-Review · Area_Chair_vhuy · 2025-03-22

**Recommendation:** Accept (Poster)
**Confidence:** 5

**Metareview:**

This paper introduces a novel joint supervised (SL) and self-supervised (SSL) learning framework for MRI reconstruction. The proposed approach effectively combines proxy fully sampled datasets with subsampled target-domain data, leveraging supervised training on proxy datasets while employing self-supervised learning on the target domain where fully sampled data is unavailable. The authors provide rigorous mathematical formulation of their method and validate it through thorough experimental analysis on two benchmark datasets, fastMRI and CMRxRecon 2023. Their results demonstrate that the hybrid training strategy enhances reconstruction performance, with fully supervised training serving as an upper-bound benchmark.

 While initial reviews highlighted certain concerns regarding the presentation of results and methodological clarity, the authors thoughtfully addressed these points during the rebuttal phase. These revisions led to an improvement in reviewers' evaluations. Overall, the work represents a meaningful contribution to the field, balancing simple but meaningful technical innovation with some improvement in performance.